# Sequential Controlled Langevin Diffusions

**Junhua Chen**[*,†,1], **Lorenz Richter**[*,2,3], **Julius Berner**[*,†,4], **Denis Blessing**[*,5],
**Gerhard Neumann**[5], **Anima Anandkumar**[6]
[1]University of Cambridge, [2]Zuse Institute Berlin, [3]dida Datenschmiede GmbH,
[4]NVIDIA, [5]Karlsruhe Institute of Technology, [6]California Institute of Technology

## Abstract

An effective approach for sampling from unnormalized densities is based on the idea of gradually transporting samples from an easy prior to the complicated target distribution. Two popular methods are (1) Sequential Monte Carlo (SMC), where the transport is performed through successive annealed densities via prescribed Markov chains and resampling steps, and (2) recently developed diffusion-based sampling methods, where a learned dynamical transport is used. Despite the common goal, both approaches have different, often complementary, advantages and drawbacks. The resampling steps in SMC allow focusing on promising regions of the space, often leading to robust performance. While the algorithm enjoys asymptotic guarantees, the lack of flexible, learnable transitions can lead to slow convergence. On the other hand, diffusion-based samplers are learned and can potentially better adapt themselves to the target at hand, yet often suffer from training instabilities. In this work, we present a principled framework for combining SMC with diffusion-based samplers by viewing both methods in continuous time and considering measures on path space. This culminates in the new *Sequential Controlled Langevin Diffusion* (SCLD) sampling method, which is able to utilize the benefits of both methods and reaches improved performance on multiple benchmark problems, in many cases using only 10% of the training budget of previous diffusion-based samplers.

## 1 Introduction

We consider the task of sampling from densities of the form

$$p_{\text{target}} = \frac{\rho_{\text{target}}}{Z} \quad \text{with} \quad Z := \int_{\mathbb{R}^d} \rho_{\text{target}}(x)\mathrm{d}x, \tag{1}$$

where $\rho_{\text{target}} \in C(\mathbb{R}^d, \mathbb{R}_{\geq 0})$ can be evaluated pointwise, but the normalizing constant $Z$ is typically intractable. This task is of great practical interest, with numerous applications in the natural sciences (Zhang et al., 2023b), for instance, for Boltzmann distributions in molecular dynamics or lattice field theory in quantum physics, as well as posterior sampling in Bayesian statistics (Gelman et al., 2013).

**Sampling problems vs. generative modeling.** The sampling problem poses unique challenges not found in other areas of probabilistic modeling. For instance, while both generative modeling and sampling involve approximating a target distribution $p_{\text{target}}$, they differ fundamentally in terms of the information available. In generative modeling, one has access to samples $X \sim p_{\text{target}}$, whereas in sampling, we only have access to a pointwise oracle $\rho_{\text{target}}$ (and, potentially, its pointwise gradients) and no samples. This distinction introduces obstacles for the sampling problem that do not exist in generative modeling. For example, a key challenge in modeling a distribution is identifying its regions of high probability, or *modes*. When samples are available, they can directly reveal the locations of these modes. In their absence, however, the sampling algorithm must include an exploration strategy to discover them and identify their shape. This exploration becomes exponentially more difficult as the dimensionality of the state space increases, making the sampling problem challenging even in moderate dimensions (e.g., $10 - 50$).

**Dynamical Measure Transport** A general idea to approach the sampling problem is to draw particles from an easy prior distribution and gradually move them toward the complicated target (sometimes termed *dynamical measure transport*). In this work, we focus on two popular paradigms:

---

*Equal contribution.    †Work partially done at the California Institute of Technology.

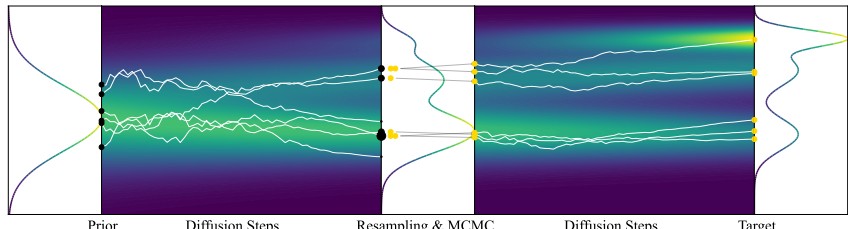

Figure 1: Illustration of our SCLD algorithm, which combines controlled Langevin diffusions with Sequential Monte Carlo methods. The goal is to sample from a target distribution by learning a stochastic evolution (diffusion steps) that starts from a tractable prior and evolves along a prescribed annealed density to the target. We do not have access to samples from the target distribution but can only evaluate its density up to a normalizing factor. At intermediate timesteps, we resample according to the importance weights of each subtrajectory (black dots) and use MCMC steps for additional refinement (yellow dots).

- In *Annealed Importance Sampling* (AIS) (Neal, 2001) and its extension *Sequential Monte Carlo* (SMC) (Chopin, 2002; Del Moral et al., 2006) particles are successively updated and reweighted, as to approach relevant regions in space, targeting an annealed sequence of intermediate distributions. This procedure is typically formulated in discrete time and does not require learning.
- In *diffusion-based sampling* (Richter & Berner, 2024; Vargas et al., 2024) the idea is to learn a drift of a stochastic differential equation (SDE) to transport the samples from the prior to the desired target, typically formulated in continuous time. The absence of samples means that data-driven approaches such as for generative modeling (Song et al., 2021) are not possible, and training is instead done via variational inference, gaining information through evaluations of $\rho_{\text{target}}$.

Each paradigm brings its own advantages and drawbacks. Traditional SMC methods rely on pre-defined rules for particle updates, such as *Markov Chain Monte Carlo* (MCMC) and resampling methods, which help to direct computational effort onto promising regions of the space and enjoy asymptotic guarantees. While they do not require learning, the employed MCMC methods can, in many cases, exhibit slow convergence to the target (Del Moral et al., 2006). Diffusion-based samplers, on the other hand, require a training phase, which enables them to automatically adapt to the given target. However, training can take significant time and often suffers from numerical instabilities as well as mode collapse (Richter & Berner, 2024).

**Sequential Controlled Langevin Diffusions.** In this work, we show that the two methods can complement each other. SMC can benefit from the flexible nature of the learnable transitions, and resampling and MCMC can help diffusion-based samplers converge faster and counteract numerical stability issues arising, for instance, from outlier particles. Motivated by this, we identify a principled and general framework to unify the two methods, culminating in our *Sequential Controlled Langevin Diffusion* (SCLD) algorithm, which alternates between SMC and diffusion steps as illustrated in Figure 1. In addition, we devise a family of loss functions that enables end-to-end training (i.e., for which the algorithm used during inference can be directly optimized). This becomes possible by viewing both methods in continuous time and considering measures of the underlying SDEs on the path space. Overall, Our contributions can be summarized as follows:

- Taking the continuous-time perspective, we can rigorously connect and unify SMC and diffusion-based sampling by performing importance sampling in path space.
- The principled framework of path space measures allows us to readily propose suitable loss functions, which allow for off-policy training with replay buffers and provably scale better to high dimensions than previously used losses.
- Building on those connections, we propose our new sampling method *Sequential Controlled Langevin Diffusion* (SCLD) as a special case of our framework.
- We show that our method achieves competitive performance on 11 real-world and synthetic examples, improving over other baseline methods in almost every task, and in many cases only using $10\%$ of the training budget. In two tasks based on robotics control, our method is the only one to approximately recover the true distribution.

## 1.1 RELATED WORK

We present an extensive comparison to related works in Appendix A.1. To summarize, our proposed SCLD sampler relies on three crucial building blocks:

Table 1: Comparison of different methods (see Appendix A.1 for details). By *discretization-flexible*, we describe the fact that we can include resampling and MCMC steps at arbitrary times. *Finite-time convergence* refers to the property that the target distribution can (theoretically, in the optimum) be reached in finite time. We note that *stochastic transitions* allow omitting or reducing (costly) MCMC steps in learned SMC methods.

|  | Traditional SMC | CMCD | CRAFT | PDDS | SCLD (ours) |
|---|---|---|---|---|---|
| Learned Transition | ✗ (MCMC) | ✓ (Neural SDE) | ✓ (Neural ODE) | ✓ (Neural SDE) | ✓ (Neural SDE) |
| Stochastic Transition | ✓ | ✓ | ✗ | ✓ | ✓ |
| End-to-end Training | - | ✓ | ✓ (needs importance weights) | ✗ (alternating) | ✓ (incl. hyperparameters) |
| Particle Method | ✓ | ✗ | ✓ | ✓ | ✓ |
| Discretization-flexible | ✓ | ✓ | ✗ | ✓ (in theory) | ✓ |
| Finite-time Convergence | ✗ | ✓ | ✓ | ✓ | ✓ |

**Sequential Monte Carlo (SMC).** SMC methods (Chopin, 2002; Del Moral et al., 2006) describe a general methodology to sample sequentially from a sequence of annealed distributions, using transition kernels (typically based on MCMC) and resampling steps. To mitigate drawbacks such as long mixing times and tedious tuning, previous works proposed to learn the kernels (Phillips et al., 2024; Matthews et al., 2022; Arbel et al., 2021). However, prior training objectives suffer from various shortcomings, either requiring importance sampling with potentially high variance, exhibiting bias, or relying on alternating methods that preclude end-to-end training (see also Table 1). Further, they need to place restrictions on their parameterizations or suffer from unfavorable computational costs. In particular, approaches with deterministic transitions, such as normalizing flows, require computations of divergences or Jacobian determinants, and MCMC steps to recover sample diversity after resampling. This is not needed for methods based on stochastic transitions like our method SCLD.

**Diffusion-based samplers on subtrajectories.** To overcome these shortcomings and flexibly parameterize the transition kernels, we draw ideas from recent work on controlled SDEs for sampling problems (Zhang et al., 2023a; Richter & Berner, 2024). This can be done by partitioning the SDE trajectories in time. However, to compute importance weights (in path space), which are necessary for resampling as well as for MCMC kernels in SMC, the SDE marginals after each subtrajectory need to be known. To cope with that, we identify the recently proposed *Controlled Monte Carlo Diffusions* (CMCD) (Vargas et al., 2024) as a suitable framework since it allows us to define a prescribed (and therefore known) target evolution of the SDE marginals. Building upon this, we develop an extension of SMC to continuous time, where resampling (and, optionally, MCMC steps) can be employed at arbitrary times.

**Log-variance loss.** However, the subtrajectories and discrete resampling steps make optimization challenging. Previous methods either relied on alternating schemes or approaches based on the reverse KL divergence and importance sampling, known to suffer from mode collapse and potentially high variance. We show that the *log-variance loss* (Nüsken & Richter, 2021) offers a way to obtain a principled, efficient, and low-variance objective such that we can optimize our sampler and parts of the hyperparameters in an end-to-end fashion using replay buffers.

## 2 Sequential controlled Langevin diffusions

We start by giving an introduction to Sequential Monte Carlo methods. However, different from previous work, our focus is on a continuous-time perspective that can be readily integrated with diffusion-based samplers.

### 2.1 A primer on Sequential Monte Carlo in continuous time

**Importance sampling (IS).** The idea of utilizing samples from a prior distribution in order to compute statistics relying on samples from a target can be motivated by importance sampling. In its simplest case, one can compute unbiased estimates w.r.t. the target distribution via

$$\mathbb{E}_{X_T \sim p_{\text{target}}}[\varphi(X_T)] = \mathbb{E}_{X_0 \sim p_{\text{prior}}}[\varphi(X_0)w(X_0)] \approx \frac{1}{K}\sum_{k=1}^{K} \varphi(X_0^{(k)})w(X_0^{(k)}), \quad (2)$$

where $\varphi \in C(\mathbb{R}^d, \mathbb{R})$ is a function of interest, the weight is defined[1] as $w := \frac{p_{\text{target}}}{p_{\text{prior}}}$, and $(X_0^{(k)})_{k=1}^{K}$ are i.i.d. samples from $p_{\text{prior}}$. Since importance sampling becomes highly inefficient if the high-

---

[1] If the normalizing constant $Z$ is not available, we can compute unnormalized weights $\widetilde{w} := \frac{\rho_{\text{target}}}{p_{\text{prior}}}$ and normalize them by their sum, leveraging the identity $Z = \mathbb{E}_{X_0 \sim p_{\text{prior}}}[\widetilde{w}(X_0)]$ (*self-normalized importance sampling*). While this introduces bias, the estimator is still consistent as $K \to \infty$ (del Moral, 2013).

probability regions of prior and target do not overlap substantially, a key idea is to gradually "transport" $X_0$ to $X_T$.

**Annealed importance sampling (AIS).** In particular, we may sequentially move particles from the prior to the target along a curve $(\pi(\cdot, t))_{t \in [0,T]}$, chosen such that $\pi(\cdot, 0) = p_{\text{prior}}$ and $\pi(\cdot, T) = p_{\text{target}}$, e.g., by linear interpolation in log-space (Dai et al., 2022). To this end, we consider two (time-dependent, forward and backward) Markov kernels $\vec{p}_{s|t}$ and $\bar{p}_{t|s}$. Given a time grid $0 = t_0 < t_1 < \cdots < t_N = T$ (also referred to as *annealing steps*), we may now sample $X_{t_0} \sim p_{\text{prior}}$ and iterate for each $n = 1, \ldots, N$:

1. Sample $X_{t_n} \sim \vec{p}_{t_n|t_{n-1}}(\cdot \mid X_{t_{n-1}})$.
2. Compute the weights $w_{t_{n-1}, t_n}(X_{t_{n-1}}, X_{t_n}) = \frac{\pi(X_{t_n}, t_n)\bar{p}_{t_{n-1}|t_n}(X_{t_{n-1}}|X_{t_n})}{\pi(X_{t_{n-1}}, t_{n-1})\vec{p}_{t_n|t_{n-1}}(X_{t_n}|X_{t_{n-1}})}$.

We can then perform importance sampling on an augmented target distribution via the weights

$$w(X_{t_0}, \ldots, X_{t_N}) := \prod_{n=1}^{N} w_{t_{n-1}, t_n}(X_{t_{n-1}}, X_{t_n}) = \frac{\bar{p}_{t_0, \ldots, t_N}(X_{t_0}, \ldots, X_{t_N})}{\vec{p}_{t_0, \ldots, t_N}(X_{t_0}, \ldots, X_{t_N})}, \quad (3)$$

where $\vec{p}_{t_0, \ldots, t_N}$ and $\bar{p}_{t_0, \ldots, t_N}$ are the joint densities of the "forward" and a corresponding "backward" operation. In particular, in analogy to (2), it holds that

$$\mathbb{E}_{X_{t_0}, \ldots, X_{t_N}}\left[\varphi(X_T) w(X_{t_0}, \ldots, X_{t_N})\right] = \mathbb{E}_{X_T \sim p_{\text{target}}}\left[\varphi(X_T)\right]. \quad (4)$$

**Resampling.** In principle, any forward and backward Markov kernels lead to an unbiased estimator of the expectation of interest, as stated in (4). In practice, however, a notorious problem with importance sampling is its potentially high variance. Specifically, the variance might increase exponentially with the dimension, sometimes termed *curse of dimensionality*, see, e.g., Chatterjee & Diaconis (2018); Hartmann & Richter (2024). To circumvent this issue, one idea is to sequentially "update" samples (also referred to as "particles") during the course of the simulation according to their weights, so as to refocus computational effort on promising particles—a procedure referred to as *resampling*. For instance, we can select only certain (relevant) samples $X_0^{(k)}$ for the estimation of the expectation in (2). To this end, let $O^{(k)}$ be a random variable with values in $\{0, \ldots, K\}$ and $\mathbb{E}[O^{(k)}|X_0^{(1)}, \ldots, X_0^{(K)}] = KW(X_0^{(k)})$, where $W(X_0^{(k)}) := w(X_0^{(k)})/\sum_{i=1}^{K} w(X_0^{(i)})$, defining how many times we select the $k$-th sample. Due to the tower property, we can then also obtain a consistent estimator of the expectation in (2) via

$$\mathbb{E}_{X_T \sim p_{\text{target}}}\left[\varphi(X_T)\right] \approx \frac{1}{K}\sum_{k=1}^{K} \varphi(X_0^{(k)}) O^{(k)}. \quad (5)$$

A common choice is to consider $O \sim \mathcal{M}_K(W(X_0^{(1)}), \ldots, W(X_0^{(K)}))$ drawn from a multinomial distribution with $K$ trials, where the normalized weights determine the event probabilities (Gordon et al., 1993). We note that with this *resampling* step, we introduce additional stochasticity. However, at the same time, it can bring statistical advantages by focusing on "relevant" samples, e.g., stabilizing effects and variance reduction (Dai et al., 2022).

**Remark 2.1** (SMC formulation in continuous vs. discrete time). We stress that, even though we evaluate our process $X$ on $N + 1$ discrete time instances, the formalism above includes time-continuous processes $(X_t)_{t \in [0,T]}$. While some transition kernels used in SMC, e.g., uncorrected Langevin kernels, can be interpreted in continuous time, SMC is typically stated for a fixed number of discrete steps. We will see in the sequel how the continuous-time formulation offers an elegant framework with certain advantages, in particular, allowing us to integrate learned SDE-based transition kernels and interleave them with resampling and MCMC steps at arbitrary times.

## 2.2 CONTROLLED SDEs AND IMPORTANCE SAMPLING IN PATH SPACE

A central question in SMC is how to choose the forward and backward transition densities $\vec{p}_{s|t}$ and $\bar{p}_{t|s}$ defined above. Clearly, when the forward and backward joint densities stated in (3) agree, we achieve perfect sampling in the sense that no corrections with importance weights are necessary. However, it is typically not possible to obtain such transitions, and thus the choice of $\vec{p}_{s|t}$ and $\bar{p}_{t|s}$ to approximate this criterion is of critical importance to the success of SMC. Whereas, traditionally, MCMC steps have been employed as the transition kernel (Dai et al., 2022), they are known to require a large number of steps to achieve approximate transportation between densities. In recent years, there has been interest in employing learned transition densities to overcome the slow convergence times of fixed MCMC kernels (Matthews et al., 2022; Phillips et al., 2024). Advancing those

---

**Algorithm 1** Sequential Controlled Langevin Diffusion (SCLD).    ▷ See Algorithm 2 for details.

---

**Require:** Annealing path $\pi$, learned control $u$, time grid $0 = t_0 < \cdots < t_N = T$
1: *Initialize:* $\overline{X}_0 := X_0^{(1:K)} \sim p_{\text{prior}}$ and $\overline{w}_0 := w_0^{(1:K)} = 1$
2: **for** $n = 1$ to $n = N$ **do**
3:    *Transport:* $\overline{X}_{[t_{n-1},t_n]} = \texttt{simulate\_SDE}\big(\overline{X}_{t_{n-1}}, u\big)$    ▷ See (6) and (19)
4:    *Compute RNDs:* $\overline{w}_{[t_{n-1},t_n]} = \frac{\mathrm{d}\overleftarrow{\mathbb{P}}_{[t_{n-1},t_n]}}{\mathrm{d}\overrightarrow{\mathbb{P}}_{[t_{n-1},t_n]}}\big(\overline{X}_{[t_{n-1},t_n]}\big)$    ▷ See (12) and (31)
5:    *Update weights:* $\overline{w}_n = \overline{w}_{n-1}\overline{w}_{[t_{n-1},t_n]}$    ▷ See (13)
6:    *Resample:* $\overline{X}_{t_n}, \overline{w}_n = \texttt{resample}\big(\overline{X}_{t_n}, \overline{w}_n\big)$    ▷ See Algorithm 5
7: **return** Samples $\overline{X}_T := X_T^{(1:K)}$ approximately from $p_{\text{target}}$

---

attempts, we will show how transition densities corresponding to SDEs yield a principled solution that, moreover, allows us to leverage recent advancements of diffusion models.

**Diffusion bridges.** To this end, let us consider the stochastic process $X^u = (X_t^u)_{t \in [0,T]}$, defined by the SDE

$$\mathrm{d}X_t^u = u(X_t^u, t)\mathrm{d}t + \sigma(t)\vec{\mathrm{d}}W_t, \qquad X_0^u \sim p_{\text{prior}}, \tag{6}$$

where $u \in C(\mathbb{R}^d \times [0,T], \mathbb{R}^d)$ is a control function, $\sigma \in C([0,T], \mathbb{R})$ the diffusion coefficient, and $W$ a standard Brownian motion. This process uniquely defines a forward transition density $\vec{p}_{s|t}$ and falls into the framework stated in Section 2.1 for any time steps $0 = t_0 < t_1 < \cdots < t_N = T$. In fact, we can leverage the ideas from CMCD (Vargas et al., 2024) and learn $u$ such that the transport happens along a prescribed density in time, i.e., such that the density $p_{X^u}(\cdot, t)$ of $X_t^u$ is equal to a prescribed target density $\pi(\cdot, t)$, connecting the prior and the target, for every $t \in [0,T]$; cf. Lemma 2.2 below. We will see that the knowledge of the marginals allows for a natural integration within SMC frameworks. Now, similar to the importance sampling framework from Section 2.1, the general idea is to exploit a time-reversed dynamics that starts in the desired target density. To be precise, we may further define a related reverse-time SDE

$$\mathrm{d}Y_t^v = v(Y_t^v, t)\mathrm{d}t + \sigma(t)\overleftarrow{\mathrm{d}}W_t, \qquad Y_T^v \sim p_{\text{target}}, \tag{7}$$

which depends on the control $v \in C(\mathbb{R}^d \times [0,T], \mathbb{R}^d)$ and where $\overleftarrow{\mathrm{d}}W_t$ denotes backward [2] integration of Brownian motion. Now, if $u$ and $v$ are learned such that $X^u$ and $Y^v$ are time-reversals of each other, then $p_{X^u} = p_{Y^v}$, i.e., the two processes transport the prior to the target and vice versa. However, in this general setting, there are infinitely many such bridging processes, all fulfilling Nelson's identity (Nelson, 1967), i.e.,

$$u - v = \sigma^2 \nabla \log p_{X^u} = \sigma^2 \nabla \log p_{Y^v}. \tag{8}$$

Since our goal is to satisfy $p_{X^u} = p_{Y^v} = \pi$, we can incorporate this constraint via the ansatz $v = u - \sigma^2 \nabla \log \pi$, leading to the SDE

$$\mathrm{d}Y_t^u = (u - \sigma^2 \nabla \log \pi)(Y_t^u, t)\mathrm{d}t + \sigma(t)\overleftarrow{\mathrm{d}}W_t, \qquad Y_T^u \sim p_{\text{target}}, \tag{9}$$

as suggested in Vargas et al. (2024), noting that the process now also depends on the control $u$. Consequently, under mild conditions, this constraint leads to a unique gradient field representing the solution $u^*$ to the time-reversal problem (Vargas et al., 2024, Proposition 3.2). We comment on more general, learnable density evolutions in Remark A.1.

**Measures in path space.** The task of learning the time-reversal can be approached via the perspective of measures on the space of continuous trajectories $C([0,T], \mathbb{R}^d)$, also called *path space*. Loosely speaking, a path space measure $\vec{\mathbb{P}} = \vec{\mathbb{P}}^{u,p_{\text{prior}}}$ of the process (6) can be thought of as the joint density $\vec{p}_{t_0,\ldots,t_N}(X_{t_0}^u, \ldots, X_{t_N}^u)$ in (3) when $N \to \infty$, i.e., evaluated along infinitely many time instances (Baldi, 2017, Corollary 11.1).

In analogy to importance sampling described in Section 2.1, we may now consider a change of measure in path space, i.e.,

$$\mathbb{E}_{X^u \sim \vec{\mathbb{P}}}\left[\varphi(X_T^u)w(X^u)\right] = \mathbb{E}_{Y^u \sim \overleftarrow{\mathbb{P}}}\left[\varphi(Y_T^u)\right] = \mathbb{E}_{x \sim p_{\text{target}}}\left[\varphi(x)\right], \tag{10}$$

---

[2] See Vargas et al. (2024, Appendix A) for details and assumptions.

where $w = \frac{\mathrm{d}\overleftarrow{\mathbb{P}}}{\mathrm{d}\overrightarrow{\mathbb{P}}}$ and $\overleftarrow{\mathbb{P}} = \overleftarrow{\mathbb{P}}^{u, p_{\text{target}}}$ is the path space measure associated to (9). Furthermore, we can formulate the time-reversal task as the minimization problem

$$u^* = \arg\min_{u \in \mathcal{U}} D\big(\overrightarrow{\mathbb{P}}^{u, p_{\text{prior}}}, \overleftarrow{\mathbb{P}}^{u, p_{\text{target}}}\big), \tag{11}$$

where $D$ is a divergence and $\mathcal{U} \subset C(\mathbb{R}^d \times [0, T], \mathbb{R}^d)$ the set of admissible controls, cf. Richter & Berner (2024). If we can bring the divergence to zero, we have indeed achieved time-reversal between the forward and backward transitions and, thus, perfect sampling. Both for (10) and typical divergences in (11), it is essential to have a tractable expression for the likelihood ratio $w$ between the measures of the forward and the reverse-time process, also called the *Radon-Nikodym derivative* (RND). This is given by the following lemma; see Vargas et al. (2024) for the proof.

**Lemma 2.2** (Likelihood ratio between path measures). *Let $\overrightarrow{\mathbb{P}}_{[s,t]}$ and $\overleftarrow{\mathbb{P}}_{[s,t]}$ be the path space measures of the solutions to the SDEs in (6) and (9) on the time interval $[s, t] \subset [0, T]$, where we assume $X_s^u \sim \pi(\cdot, s)$ and $Y_t^u \sim \pi(\cdot, t)$. Then for a generic[3] process $X$ it holds*

$$
\begin{aligned}
w_{[s,t]}(X) = \frac{\mathrm{d}\overleftarrow{\mathbb{P}}_{[s,t]}}{\mathrm{d}\overrightarrow{\mathbb{P}}_{[s,t]}}(X) = {} & \frac{\pi(X_t, t)}{\pi(X_s, s)} \exp\bigg( \int_s^t \frac{\|u\|^2 - \|u - \sigma^2 \nabla \log \pi\|^2}{2\sigma^2}(X_\tau, \tau) \mathrm{d}\tau \\
& + \int_s^t \frac{u - \sigma^2 \nabla \log \pi}{\sigma^2}(X_\tau, \tau) \cdot \overleftarrow{\mathrm{d}} X_\tau - \int_s^t \frac{u}{\sigma^2}(X_\tau, \tau) \cdot \overrightarrow{\mathrm{d}} X_\tau \bigg).
\end{aligned}
\tag{12}
$$

As can be seen from Lemma 2.2, path space measures can be readily employed for sequential algorithms that operate on the time grid that we introduced before. In particular, we may divide our trajectories $X^u$ and $Y^u$ into subtrajectories and thus our path space measure into multiple chunks. To be precise, we may write

$$w = \frac{\mathrm{d}\overleftarrow{\mathbb{P}}}{\mathrm{d}\overrightarrow{\mathbb{P}}} = \frac{\mathrm{d}\overleftarrow{\mathbb{P}}_{[t_0, t_1]}}{\mathrm{d}\overrightarrow{\mathbb{P}}_{[t_0, t_1]}} \cdots \frac{\mathrm{d}\overleftarrow{\mathbb{P}}_{[t_{N-1}, t_N]}}{\mathrm{d}\overrightarrow{\mathbb{P}}_{[t_{N-1}, t_N]}} = w_{[t_0, t_1]} \cdots w_{[t_{N-1}, t_N]}. \tag{13}$$

Different from the framework in Section 2.1, we note that Lemma 2.2 offers an explicit formula for computing the weights $w_{[t_{n-1}, t_n]}$ in continuous time. As can be seen in the importance sampling identity (10), the weights can be interpreted as correcting for a potentially imperfect time-reversal. For convenience, we state Algorithm 1 for a simplified, high-level overview of combining SMC with diffusion models and refer to Algorithm 2 in Appendix A.3 for a more detailed exposition. Further, we note that the suggested setting relates to the usual SMC algorithm (such as in Dai et al. (2022)) by taking a different forward transport step (where our Markov kernel is implemented by an SDE) and by adopting the weighting step (using the Radon-Nikodym derivative in place of the likelihood ratio). Using the target density $\pi(\cdot, t_n)$, we can also add MCMC refinements at each time $t_n$; see Section 2.4.

## 2.3 LOSS FUNCTIONS AND OFF-POLICY TRAINING

We can adapt the idea of learning the optimal control $u^*$ to our sequential setting by considering divergences on each subinterval $[t_{n-1}, t_n]$ separately, in consequence bringing losses of the form

$$\mathcal{L}(u) = \sum_{n=1}^N D\left(\overrightarrow{\mathbb{P}}_{[t_{n-1}, t_n]}^{u, \pi_{n-1}}, \overleftarrow{\mathbb{P}}_{[t_{n-1}, t_n]}^{u, \pi_n}\right), \tag{14}$$

where $\pi_n := \pi(\cdot, t_n)$. We stress that with (14) optimization can in principle be conducted globally in spite of the resampling happening sequentially. However, depending on the choice of the divergence, this comes with additional challenges.

**KL divergence.** A classical choice is the *Kullback-Leibler (KL) divergence* $D = D_{\text{KL}}$, i.e.,

$$D_{\text{KL}}\left(\overrightarrow{\mathbb{P}}_{[t_{n-1}, t_n]}^{u, \pi_{n-1}} | \overleftarrow{\mathbb{P}}_{[t_{n-1}, t_n]}^{u, \pi_n}\right) = -\mathbb{E}_{X^u \sim \overrightarrow{\mathbb{P}}_{[t_{n-1}, t_n]}^{u, \pi_{n-1}}}\left[\log\big(w_{[t_{n-1}, t_n]}(X^u)\big)\right], \tag{15}$$

where $w_{[t_{n-1}, t_n]}$ is defined as in Lemma 2.2 and the minus originates from the reciprocal importance weights in the logarithm. However, for computing the expectation we need $X_{t_{n-1}}^u \sim \pi_{n-1}$. If resampling has been employed in the previous iteration (at time $t_{n-1}$; see Algorithm 1), a potential mismatch in the expectation is automatically corrected. Alternatively, we may correct with importance sampling in path space. To this end, let $t_m$ (with $t_m < t_{n-1}$) be the last time resampling has been conducted, i.e., the last time the weights have been reset; see Algorithm 5. As suggested in

---

[3]Note that the Radon-Nikodym derivative is only defined almost surely w.r.t. $\overrightarrow{\mathbb{P}}_{[s,t]}$. In particular, it only depends on $X$ on the time interval $[s, t]$.

Matthews et al. (2022), we can then consider the importance weight $w_{[t_m, t_{n-1}]}$ and compute

$$D_{\mathrm{KL}}\left(\vec{\mathbb{P}}^{u,\pi_{n-1}}_{[t_{n-1},t_n]} | \vec{\mathbb{P}}^{u,\pi_n}_{[t_{n-1},t_n]}\right) = -\mathbb{E}_{X^u \sim \vec{\mathbb{P}}^{u,\pi_m}_{[t_m,t_n]}}\left[\log\left(w_{[t_{n-1},t_n]}(X^u)\right) w_{[t_m,t_{n-1}]}(X^u)\right], \quad (16)$$

for which $X^u_{t_{n-1}}$ does not need to be distributed according to $\pi_{n-1}$ anymore. However, the importance weights potentially introduce additional variance into the loss, particularly in high dimensions. This observation is stated rigorously in the following proposition, cf. Nüsken & Richter (2021, Proposition 5.7), and proved in Appendix A.2.

**Proposition 2.3** (Relative error of KL divergence). *Denote by $D_{\chi^2}$ the $\chi^2$-divergence and by $r^{(K)} := \mathrm{Var}(\widehat{D}^{(K)}_{\mathrm{KL}})^{1/2}/D_{\mathrm{KL}}$ the relative error of the Monte Carlo estimator $\widehat{D}^{(K)}_{\mathrm{KL}}$ of the KL divergence in (16) with sample size $K$. Moreover, let $t_m$ be the last resampling time and let $\vec{\mathbb{P}}^{\otimes I}_{[t_{n-1},t_n]}$ and $\overleftarrow{\mathbb{P}}^{\otimes I}_{[t_{n-1},t_n]}$ be the $I$-fold product measures of identical copies of $\vec{\mathbb{P}}_{[t_{n-1},t_n]}$ and $\overleftarrow{\mathbb{P}}_{[t_{n-1},t_n]}$, respectively. Then there exists a constant $c > 0$, such that for any $I \geq 2$ it holds that*

$$r^{(K)}\left(\vec{\mathbb{P}}^{\otimes I}_{[t_{n-1},t_n]} | \overleftarrow{\mathbb{P}}^{\otimes I}_{[t_{n-1},t_n]}\right) \geq c\left(D_{\chi^2}\left(\overleftarrow{\mathbb{P}}_{[t_m,t_{n-1}]} | \vec{\mathbb{P}}_{[t_m,t_{n-1}]}\right) + 1\right)^{I/2}. \quad (17)$$

Given a path measure $\vec{\mathbb{P}}$ of a $D$-dimensional process, we note that $\vec{\mathbb{P}}^{\otimes I}$ is a measure on the product space $\bigotimes_{i=1}^{I} C([0,T], \mathbb{R}^D) \simeq C([0,T], \mathbb{R}^{ID})$. In particular, for $D = 1$ (corresponding to independent components), we can clearly identify $d = I$ as the dimension of the considered problem. This means that the relative error of the estimator of the KL divergence (16) is expected to scale exponentially in the dimension, which is illustrated in Figure 11 in Appendix A.6.10. As shown in Nüsken & Richter (2021), the *log-variance (LV) divergence* does not exhibit this unfavorable property.

**LV divergence and off-policy training.** An alternative divergence can be defined by

$$D^{\mathbb{Q}}_{\mathrm{LV}}\left(\vec{\mathbb{P}}^{u,\pi_{n-1}}_{[t_{n-1},t_n]} | \overleftarrow{\mathbb{P}}^{u,\pi_n}_{[t_{n-1},t_n]}\right) = \mathrm{Var}_{X \sim \mathbb{Q}}\left[\log\left(w_{[t_{n-1},t_n]}(X)\right)\right] \quad (18)$$

which, in fact, is a family of divergences parametrized by a reference measure $\mathbb{Q} = \vec{\mathbb{P}}^{\widetilde{u},\widetilde{\pi}_{n-1}}_{[t_{n-1},t_n]}$ that can be chosen with arbitrary controls $\widetilde{u}$ and initial distributions $\widetilde{\pi}_{n-1}$ (also called *off-policy training*, see Remark A.2 for details and connections to reinforcement learning). In particular, we do not need $X_{t_{n-1}} \sim \pi_{n-1}$ anymore, and thus reweighting such as in (16) is not necessary, irrespective of the fact that resampling at time $t_n$ might not have been conducted. We summarize the training procedure for both divergences in Algorithm 3 and present details in Appendix A.3.

## 2.4 ALGORITHMIC REFINEMENTS AND IMPLEMENTATIONAL DETAILS

In this section, we turn our theoretical considerations from Sections 2.1 to 2.3 into implementable algorithms. We collate these changes in Algorithm 2 in Appendix A.3, representing a practical version of Algorithm 1.

**Loss Function.** We focus on the log-variance divergence in the sequel and refer to Appendix A.6.10 for a comparison to the KL divergence. We choose $\widetilde{u} = u$ (or previous versions when using a buffer, see "replay buffers" below) and simulate $X$ in (18) starting from the prior, so $\widetilde{\pi}_n$ corresponds to the SDE marginal. However, since we do not take gradients w.r.t. the control $\widetilde{u}$ of the reference measures, we detach the trajectory $X$, in line with Richter & Berner (2024). In particular, we do not need to differentiate through the SDE integrator.

**Time discretization.** In practice, we choose $N$ equidistant resampling times, i.e. $t_n - t_{n-1} = \tau$, for every $n \in \{1, \ldots, N\}$, where the number of subtrajectories $N$ may change across applications. We discretize the SDE (7) via the Euler-Maruyama scheme, containing $L$ evenly spaced steps per subtrajectory, i.e.,

$$\widehat{X}^u_i = \widehat{X}^u_{i-1} + u(\widehat{X}^u_{i-1}, (i-1)h)h + \sigma((i-1)h)\sqrt{h}\xi_i, \quad \xi_i \sim \mathcal{N}(0, \mathrm{Id}), \quad (19)$$

for $i \in \{1, \ldots, NL\}$ with $h = \tau/L$. We refer to (31) in Appendix A.3 for the resulting discretization of the Radon-Nikodym derivative from Lemma 2.2 for computing the importance weights $w_{[t_{n-1},t_n]}$.

**Annealing path.** For the prescribed density curve $\pi$ we consider

$$\pi(x,t) \propto p_{\mathrm{prior}}(x)^{1-\beta(t)} \rho_{\mathrm{target}}(x)^{\beta(t)}, \quad (20)$$

Table 2: Comparison of different methods in terms of ELBOs, i.e., lower bounds on the log-normalization constant $\log Z$. We use this metric for all tasks where we do not have access to groundtruth metrics. We report NA if all considered hyperparameter choices diverged.

| ELBO ($\uparrow$) | Seeds (26d) | Sonar (61d) | Credit (25d) | Brownian (32d) | LGCP (1600d) |
|---|---|---|---|---|---|
| **SMC** | $-74.63_{\pm 0.14}$ | $-111.50_{\pm 0.96}$ | $-589.82_{\pm 5.72}$ | $-2.21_{\pm 0.53}$ | $385.75_{\pm 7.65}$ |
| **SMC-ESS** | $-74.07_{\pm 0.60}$ | $-109.10_{\pm 0.17}$ | $-505.57_{\pm 0.18}$ | $0.49_{\pm 0.19}$ | $\mathbf{497.85_{\pm 0.11}}$ |
| **SMC-FC** | $-74.07_{\pm 0.02}$ | $-108.93_{\pm 0.02}$ | $-505.30_{\pm 0.02}$ | $-1.91_{\pm 0.04}$ | $-878.10_{\pm 2.20}$ |
| **CRAFT** | $-73.75_{\pm 0.02}$ | $-108.97_{\pm 0.16}$ | $-518.25_{\pm 0.52}$ | $0.90_{\pm 0.10}$ | $485.87_{\pm 0.37}$ |
| **DDS** | $-75.21_{\pm 0.21}$ | $-121.22_{\pm 5.99}$ | $-514.74_{\pm 1.22}$ | $0.56_{\pm 0.23}$ | NA |
| **PIS** | $-88.92_{\pm 2.05}$ | $-142.87_{\pm 3.29}$ | $-846.57_{\pm 2.42}$ | NA | $479.54_{\pm 0.40}$ |
| **CMCD-KL** | $-73.51_{\pm 0.01}$ | $-109.09_{\pm 0.01}$ | $-507.23_{\pm 6.40}$ | $0.86_{\pm 0.01}$ | $478.75_{\pm 0.34}$ |
| **CMCD-LV** | $-73.67_{\pm 0.01}$ | $-109.50_{\pm 0.03}$ | $-504.90_{\pm 0.02}$ | $0.54_{\pm 0.03}$ | $472.79_{\pm 0.44}$ |
| **SCLD (ours)** | $\mathbf{-73.45_{\pm 0.01}}$ | $\mathbf{-108.17_{\pm 0.25}}$ | $\mathbf{-504.46_{\pm 0.09}}$ | $\mathbf{1.00_{\pm 0.18}}$ | $486.77_{\pm 0.70}$ |

where $\beta \colon [0, T] \to [0, 1]$ is a monotonically increasing function fulfilling $\beta(0) = 0$ and $\beta(T) = 1$. We choose to learn the function $\beta$ to attain a smoother transition; see (35) and Appendix A.6.5.

**Resampling.** There is a wealth of literature (Webber, 2019; Doucet et al., 2001; Douc & Cappé, 2005) regarding designing SMC resampling schemes. However, for a fair comparison to CRAFT (Matthews et al., 2022), we utilize the common *multinomial* resampling scheme. Resampling can, however, reduce particle diversity by introducing identical particles in its output. As such, it is common to trigger resampling at a time $t_n$ only when the *Effective Sample Size* (ESS), a measure of particle quality defined by $\text{ESS} = \frac{\left(\sum_{k=1}^{K} w_n^{(k)}\right)^2}{\sum_{k=1}^{K} (w_n^{(k)})^2}$, is below a certain threshold, where $w_n$ are the importance weights at time $t_n$ (as in Algorithm 2). In line with prior works (Matthews et al., 2022; Phillips et al., 2024), we pick the threshold to be $0.3K$ where $K$ is the number of particles.

**MCMC refinements.** In order to cope with sub-optimal controls $u$ during the course of optimization, we add some MCMC refinement steps after each subtrajectory at time $t_n$, using a Markov kernel with invariant measure $\pi(\cdot, t_n)$. In line with Matthews et al. (2022), after every subtrajectory, we use one Hamiltonian Monte Carlo (HMC) step with 10 leapfrog steps.

**Replay buffers.** Replay buffers are known to prevent mode collapse and improve sample efficiency for sampling tasks (Vemgal et al., 2023; Midgley et al., 2022; Sendera et al., 2024). As such, we utilize a prioritized replay buffer during training time. At a high level, we maintain a fixed-size rolling cache of paths generated by previous versions of the *policy*, i.e., learned control $u$. For the gradient updates, we then take half of the samples from the current policy and the other from the buffer using Radon-Nikodym derivatives as weights for prioritization, see Algorithm 4 in Appendix A.3 for details. We note that this procedure is easily feasible with the log-variance divergence since this divergence does not rely on an evaluation along the current policy (see Section 2.3).

## 3 EXPERIMENTS

We empirically demonstrate the performance of the proposed SCLD sampler on a wide variety of sampling benchmarks.[4] We consider a combination of practical and synthetic examples taken from Blessing et al. (2024), the full descriptions of which are contained in Appendix A.4:

- **Examples from Bayesian statistics:** The Seeds, Sonar, Credit, Brownian, and LGCP tasks.

- **Synthetic targets:** A 40-mode Gaussian mixture model in $50d$ (GMM40), a 32-mode Many-Well task (MW54) in $5d$, the popular $10d$ Funnel benchmark, and a $50d$ Student mixture model (MoS). Many of these are in relatively high dimensions and with many well-separated modes.

- **The Robot1 and Robot4 tasks:** Inspired by robotics control problems, these synthetic 10-dimensional targets model the distribution over the configurations of a 10-joint robotic arm in the plane. They have multiple well-separated and sharp modes.

As baselines, we consider a representative selection of related sampling methods and refer to Appendix A.1 for descriptions. We study two metrics used frequently by previous works, such as in Blessing et al. (2024); Vargas et al. (2023). When groundtruth samples are available, we report the Sinkhorn distance (an optimal transport distance) to a set of generated samples (Cuturi, 2013), and otherwise consider the ELBO metric (i.e., a lower bound on $\log Z$).

---

[4]Our code can be found at `https://github.com/anonymous3141/SCLD`.

Table 3: Comparison of different methods in terms of Sinkhorn distances. We present all tasks where we have access to samples for the evaluation. We report NA if all considered hyperparameter choices diverged.

| Sinkhorn ($\downarrow$) | Funnel (10d) | MW54 (5d) | Robot1 (10d) | Robot4 (10d) | GMM40 (50d) | MoS (50d) |
|---|---|---|---|---|---|---|
| **SMC** | $149.35_{\pm4.73}$ | $20.71_{\pm5.33}$ | $24.02_{\pm1.06}$ | $24.08_{\pm0.26}$ | $46370.34_{\pm137.79}$ | $3297.28_{\pm2184.54}$ |
| **SMC-ESS** | $\mathbf{117.48_{\pm9.70}}$ | $1.11_{\pm0.15}$ | $1.82_{\pm0.50}$ | $2.11_{\pm0.31}$ | $24240.68_{\pm50.52}$ | $1477.04_{\pm133.80}$ |
| **SMC-FC** | $211.43_{\pm30.08}$ | $2.03_{\pm0.17}$ | $0.37_{\pm0.08}$ | $1.23_{\pm0.02}$ | $39018.27_{\pm159.32}$ | $3200.10_{\pm95.35}$ |
| **CRAFT** | $133.42_{\pm1.04}$ | $11.47_{\pm0.90}$ | $2.92_{\pm0.01}$ | $4.14_{\pm0.50}$ | $28960.70_{\pm354.89}$ | $1918.14_{\pm108.22}$ |
| **DDS** | $142.89_{\pm9.55}$ | $0.63_{\pm0.24}$ | $11.44_{\pm12.50}$ | $5.38_{\pm2.44}$ | $5435.18_{\pm172.20}$ | $2154.88_{\pm3.86}$ |
| **PIS** | NA | $\mathbf{0.42_{\pm0.01}}$ | $1.54_{\pm0.72}$ | $2.02_{\pm0.36}$ | $10405.75_{\pm69.41}$ | $2113.17_{\pm31.17}$ |
| **CMCD-KL** | $124.89_{\pm8.95}$ | $0.57_{\pm0.05}$ | $3.71_{\pm1.00}$ | $2.62_{\pm0.41}$ | $22132.28_{\pm595.18}$ | $1848.89_{\pm532.56}$ |
| **CMCD-LV** | $139.07_{\pm9.35}$ | $0.51_{\pm0.08}$ | $28.49_{\pm0.07}$ | $27.00_{\pm0.07}$ | $4258.57_{\pm737.15}$ | $1945.71_{\pm48.79}$ |
| **SCLD (ours)** | $134.23_{\pm8.39}$ | $0.44_{\pm0.06}$ | $\mathbf{0.31_{\pm0.04}}$ | $\mathbf{0.40_{\pm0.01}}$ | $\mathbf{3787.73_{\pm249.75}}$ | $\mathbf{656.10_{\pm88.97}}$ |

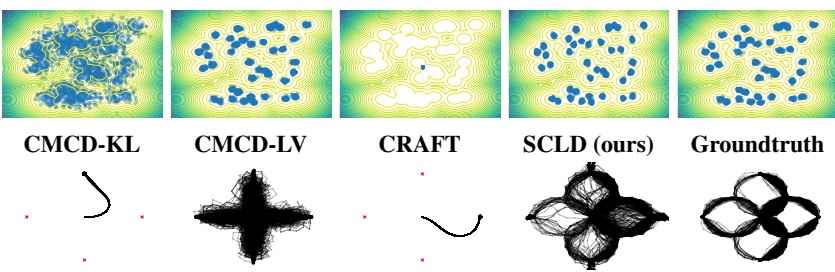

**CMCD-KL**      **CMCD-LV**      **CRAFT**      **SCLD (ours)**      **Groundtruth**

Figure 2: Samples from our considered methods and the groundtruth for the GMM40 (50d) (top) and Robot4 (10d) (bottom) tasks. Our SCLD method accurately finds all modes and avoids low probablity regions.

We took great care to ensure the fairness of our experiments and refer the reader to Appendix A.5 for full experimental and reproducibility details and to Blessing et al. (2024) for a discussion on benchmarking samplers. We also include numerous additional experiments and metrics in the appendices, such as ablation studies in Appendices A.6.1 and A.6.2, runtime information in Appendix A.6.3, a study on $\log Z$ estimation in Appendix A.6.4, the effect of learning priors by variational inference in Appendix A.6.6, a comparison to PDDS in Appendix A.6.7, and a comparison of KL and LV training in Appendix A.6.10.

## 3.1 RESULTS

Our SCLD method exhibits strong performance on both ELBO and Sinkhorn benchmarks (Tables 2 and 3). Indeed, among all tasks except Funnel, we are able to achieve the top performance or come a close second when measuring performance by Sinkhorn distances (when it is available). For ELBO estimation, SCLD can utilize a large number of resampling steps to attain the strongest performances in all but one task. In particular, SCLD can surpass the outcomes of CMCD-KL and CMCD-LV with 40000 gradient steps using only 3000 steps. In the following, we comment on different aspects.

**Avoiding mode collapse.** We visualize the samples for GMM40 and the Robot4 task in Figure 2. For GMM40, we plot the first two dimensions of samples against the true marginal distribution. In all attempted hyperparameter settings, we found that CRAFT suffers from mode collapse (see also Appendix A.6.4) and that CMCD-KL gradually collapses to a few modes, covering low probability regions. CMCD-LV and SCLD perform much better, and indeed the samples from SCLD are virtually indistinguishable from the groundtruth. For Robot4, we visualize the sampled robot arm positions. Observe that for the Robot4 task, CMCD-KL and CRAFT both collapse onto 1 mode. CMCD-LV does not experience mode collapse but nevertheless does not sample accurately for any mode. Only our SCLD Method is able to identify and sample relatively precisely from all 8 modes.

**Improved convergence properties of SCLD.** We found that the SCLD algorithm demonstrates superior convergence properties. As SCLD is effectively initialized as an SMC sampler and is trained to improve upon it, we expect a good initial performance even before training and, thus, an improved starting point for optimization. As visualized in Figure 3, SCLD consistently attains better ELBOs for any given training time budget on all tasks when compared to CMCD-KL and CMCD-LV. While in some cases SCLD is initially worse than CRAFT, it always manages to catch up quickly and surpasses it. SCLD and CMCD steps require similar amounts of time for these tasks (see Appendix A.6.3), and thus SCLD offers a 10-fold decrease in training time as well as iteration count compared to CMCD. See Appendix A.6.9 for an alternative visualization.

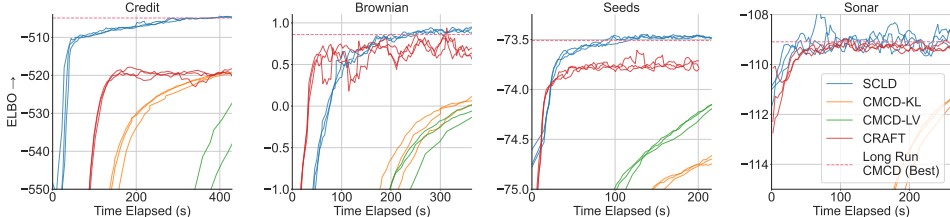

Figure 3: ELBOs during training for several tasks. We visualize the ELBO estimates attained by 4 methods as a function of the training time elapsed (until SCLD finished after 3000 iterations), running 3 seeds for each task. We mark the long run CMCD ELBOs (best out of KL and LV loss), corresponding to running for 40000 gradient steps as for the main table. Methods leveraging Sequential Monte Carlo (SCLD and CRAFT) generally exhibit improved convergence speed, but whereas CRAFT plateaus quickly, our SCLD method often achieves state-of-the-art performance in about 5 minutes.

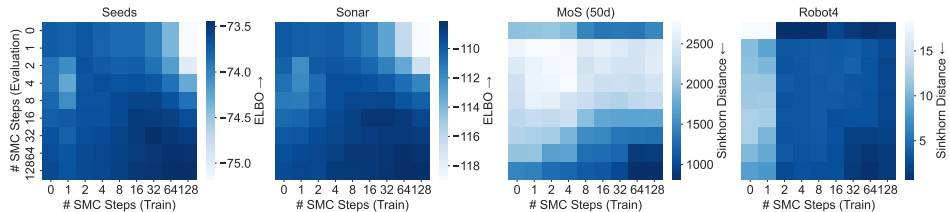

Figure 4: Performance of SCLD for different numbers of SMC steps at training and evaluation time for several tasks. Better results are shaded darker. We note that taking "zero" SMC steps corresponds to the CMCD method. Using more SMC steps has generally a beneficial effect during training. Our method allows us to select a different number at training and during inference.

**Choice of number of SMC steps.** Here, we study the effect of varying the number $N$ of subtrajectories used in the SCLD sampler, i.e., SMC steps where we apply resampling (if ESS is lower than the threshold) and MCMC steps, and offer practical advice on choosing this value. For this study, we fix the number of gradient steps for training to 8000 but otherwise retain the same experimental design. The results are illustrated in Figure 4, where we visualize the relevant metric for four tasks and demonstrate the effect of varying the number of SMC steps used at training and evaluation. For most tasks, we found it advantageous to use as many SMC steps as possible at both training and evaluation time. Particularly for the Seeds and Sonar targets, the outcomes look strikingly similar. For these tasks, it is also shown that using a smaller number of SMC steps at training or even only adding SMC steps at evaluation already improves upon stand-alone diffusion-based samplers.

While it is well known that resampling can potentially lead to mode collapse and loss of sample diversity on highly multimodal tasks (Doucet et al., 2001), we found that even for such tasks, re-sampling, when used sparingly, was still beneficial during training. This is clearly reflected in the multimodal Robot4 task, where using SMC steps at training significantly improves sample quality. In line with the previous paragraph, this suggests that our SCLD training setup can help improve training convergence. Informed by our observations, we opt to use 4 subtrajectories only at training for all synthetic tasks except Funnel and MoS and, for all other tasks, we utilize 128 subtrajectories at both training and evaluation time for the main experiments. These choices, while not necessarily optimal, are robust and work well across our diverse set of benchmarks.

## 4 CONCLUSION

We have developed a framework for combining diffusion-based samplers with Sequential Monte Carlo algorithms and propose simple yet effective methods for training. Our framework culminates in a novel sampler, termed *Sequential Controlled Langevin Diffusion* (SCLD), in principle offering a great amount of design freedom. In particular, SCLD allows for accelerated training, flexible parameterizations, end-to-end training with prioritized replay buffers, and injection of resampling and MCMC steps at arbitrary times in the generative process. We provide careful ablation studies of our design choices and empirically show state-of-the-art performance on a diverse range of benchmarks.

ACKNOWLEDGEMENTS

We thank Francisco Vargas, Lee Cheuk-Kit, and Dinghuai Zhang for very helpful discussions. J.C. was supported by a Summer Undergraduate Research Fellowship at the California Institute of Technology. D.B. is supported by funding from the pilot program Core Informatics of the Helmholtz Association (HGF). J.B. acknowledges support from the Wally Baer and Jeri Weiss Postdoctoral Fellowship. A.A. is supported in part by Bren endowed chair, ONR (MURI grant N00014-23-1-2654), and the AI2050 senior fellow program at Schmidt Sciences. The research of L.R. was partially funded by Deutsche Forschungsgemeinschaft (DFG) through the grant CRC 1114 "Scaling Cascades in Complex Systems" (project A05, project number 235221301). We thank the anonymous reviewers for their careful reading of our manuscript and their insightful comments and suggestions.

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

## A APPENDIX

CONTENTS

### A.1 RELATED WORKS

Adding to Section 1.1, this section provides additional related works.

**SMC.** SMC methods (Chopin, 2002; Del Moral et al., 2006) describe a general methodology to sample sequentially from a sequence of (annealed) distributions. They rely on forward and backward kernels in order to move from one distribution to another and leverage resampling steps in between. Popular choices for the kernels include MCMC. However, while enjoying theoretical guarantees, they suffer from drawbacks such as long mixing times and tedious tuning (Dai et al., 2022).

**SMC with learned kernels.** To make the transition kernels more flexible and reduce the amount of manual tuning, previous approaches have been proposed to learn them (Wu et al., 2020; Geffner & Domke, 2021). Combinations with SMC include the works by Bernton et al. (2019); Heng et al. (2017). While they propose learned SMC transitions, they do not utilize neural networks (partially due to tractability issues). Bernton et al. (2019) build on the prior work of Heng et al. (2017), which uses ideas from optimal control to iteratively modify the prior distribution and transition kernels through an approximate dynamic programming approach. However, this requires the prior distribution to be conjugate with respect to the policy of the underlying optimal control problem, among other drawbacks discussed in Bernton et al. (2019). The latter work, in turn, proposes the *Sequential Schrödinger Bridge Sampler* (SSB), which produces a trained SMC sampler by applying sequential approximate iterative proportional fitting (IPF) to learn the forward and backward kernels.

Whereas the paper works in discrete time, we take a continuous time perspective and, in doing so, obtain a family of simpler, unbiased training procedures, as well as reveal additional design choices like the ability to choose the integrator. We also note that our objective is fundamentally different from IPF and, in particular, yields a different solution for a finite numbers of steps (see Vargas et al. (2024, Proposition 3.4)).

Methods combining SMC with neural networks include *Annealed Flow Transport Monte Carlo* (AFT) (Arbel et al., 2021), as well as its improved version *Continual Repeated Annealed Flow Transport Monte Carlo* (CRAFT) (Matthews et al., 2022). Those works use normalizing flows to transition between adjacent annealing steps. While achieving improved performance, the deterministic nature of the transitions requires MCMC steps after the resampling steps to avoid particles collapsing to the same location. Moreover, the log-determinant of the Jacobian (or divergence of the drift for continuous time) is required. To avoid costly computations in high dimensions, one either needs to place architectural restrictions on the architecture or require the use of noisy estimators (such as Hutchinson's trace estimator (Hutchinson, 1989) for the divergence). We remark that there is also a series of works that combines normalizing flows with MCMC methods (Midgley et al., 2022; Gabrié et al., 2021; 2022; Hagemann et al., 2023).

**Diffusion-based samplers.** Works on diffusion-based samplers such as *Path Integral Samplers* (PIS), *Denoising Diffusion Samplers (DDS)*, *Time-reversed Diffusion Samplers (DIS)*, and others introduced by Zhang & Chen (2022); Berner et al. (2024); Vargas et al. (2023; 2024); Sendera et al. (2024); Sun et al. (2024) have focused on transporting a prior to the target distribution using controlled stochastic differential equations (SDEs), where the control is learned by minimizing suitable divergences between induced measures on the SDE trajectories; see the framework described in Section 2.2. As a historical note, some of the ideas of diffusion sampling were anticipated in earlier works such as Vaikuntanathan & Jarzynski (2008). In this work, we aim to harness their flexibility together with the power of SMC. Orthogonal to our work, techniques from diffusion models have been employed to approximate the extended target distribution needed in AIS methods (Doucet et al., 2022; Geffner & Domke, 2022).

**Subtrajectories.** In our work, we utilize the idea of dividing a path measure into sequential sections. This bears resemblance to the concept of subtrajectories as introduced in a discrete-time setting in the context of GFlownets (Zhang et al., 2023a; Madan et al., 2023), and thus we will also use this term. While conceptually similar, the latter work only proposed subtrajectories as an alternative training loss, whereas we use them to facilitate integration with SMC methods. Additionally, their formulation requires learning the evolution of the SDE marginals, whereas we adapt recent *Controlled Monte Carlo Diffusions* (CMCD) (Vargas et al., 2024) to get rid of this requirement.

**SMC with diffusion-based samplers.** To the best of our knowledge, the only prior work on combining diffusion-based methods with SMC steps is the *Particle Denoising Diffusion Sampler* (PDDS) (Phillips et al., 2024), where the backward kernel is chosen to be the noising diffusion and the forward kernel the approximate (learned) time-reversal. While also inspired by diffusion-based samplers, PDDS significantly differs from our approach. First, we take a more general continuous-time perspective, allowing us more freedom in design choices while still recovering the (discrete-time) setup of PDDS as a special case (i.e., where we use one Euler-Marumaya step per subtrajectory). Next, their setup requires learning potential functions and relies on automatic differentiation to compute the control, which can be unstable and challenging to optimize. Indeed, PDDS was empirically found to require variational approximations for the prior distribution to train stably, which has certain drawbacks (see Appendices A.6.6 and A.6.7). Moreover, it uses an alternating training setup that uses (approximate) samples from the partially trained model, whereas we train our model end-to-end, i.e., our setup is the same during training and inference. We empirically compare methods and discuss the impact of this difference in training methodology in Appendix A.6.7. We additionally compare different SMC-based methods in Table 1. Lastly, we note that the recent concurrent work by Albergo & Vanden-Eijnden (2025) also utilizes resampling steps as part of a learned SDE-based sampling algorithm.

**Diffusion-based generative modeling.** As outlined in our introduction, sampling problems are substantially different from problems in generative modeling, where samples from the target distribution are provided. However, many successful techniques from diffusion-based generative modeling, such as SDE integrators, noise schedules, and probability flow ODEs, can be translated to diffusion-based samplers. Loosely related to CMCD, and thus SCLD, are (entropic) *action-matching* ap-

proaches (Neklyudov et al., 2023), where the intermediate distributions are prescribed via samples as compared to (unnormalized) densities in our setting. In both settings, there exist unique gradient fields representing the optimal controls, which can be characterized as solutions to infinitesimal *Schrödinger bridge problems* at the intermediate distributions, i.e., minimizers of the kinetic energy (see Vargas et al. (2024, Proposition 3.4) and Neklyudov et al. (2023, Appendix B.3)). As described in Remark A.1, we could replace the CMCD framework with more general bridges, i.e., arbitrary, learnable density evolutions as considered in Richter & Berner (2024), at the cost of learning the (unnormalized) marginals with a separate model. A corresponding objective in generative modeling has been considered by Chen et al. (2022). While such approaches do not exhibit unique solutions, one can additionally minimize the KL divergence of the learned path measure to a reference measure, typically given by a Brownian motion, which leads to dynamic Schrödinger bridge problems (i.e., entropy-regularized optimal transport). This has been explored by, e.g., Vargas et al. (2021); De Bortoli et al. (2021); Shi et al. (2024) in the context of generative modeling, and we refer to Koshizuka & Sato (2023); Liu et al. (2022; 2023) for extensions beyond kinetic energy minimization (related to *mean-field games*).

Finally, we mention that generative modeling frameworks that allow likelihood computations can also be used for sampling problems. Specifically, one can optimize objectives from generative modeling (e.g., score-matching objectives) using approximate samples from the target distribution obtained from the partially trained model together with importance sampling based on the likelihoods of the samples. This can be viewed as a version of the *cross-entropy method* and is used, e.g., in Jing et al. (2022, Section 3.6) for diffusion models[5] and in Tong et al. (2024, Appendix C.2) for flow matching. However, a mismatch of the high-probability regions of the proposal (given by the partially trained model) and target distributions often leads to high variance in high-dimensional settings. We note that PDDS can be viewed as a very elaborate version of such an approach, counteracting the aforementioned problems by incorporating SMC steps into the proposal as well as training with a combination of target-matching and score-matching objectives. We compare to PDDS in Appendix A.6.7.

**Diffusion-based posterior sampling and stochastic optimal control.** For our considered sampling problems, we only assume minimal to no prior knowledge of the properties of the target distribution. However, for sampling from posterior distributions arising from Bayesian inference problems, one can decompose the target as $p_{\text{target}} = p_{X|Y}(\cdot, y) = \frac{p_X p_{Y|X}(y|\cdot)}{Z}$, where $y$ is a given measurement and $p_X$ and $p_{Y|X}$ are the prior and likelihood, respectively. In our Bayesian statistics tasks (see Appendix A.4), the prior $p_X$ is given by a simple, tractable distribution, and we do not incorporate knowledge about the prior into our framework.

However, for certain problems, the prior can also be more complex, e.g., in inverse problems on image, audio, or video distributions. Assuming – different from our setting – that samples from the prior $p_X$ are given, recent methods leverage *diffusion priors*, i.e., diffusion models pre-trained on $p_X$, to simplify sampling from $p_{\text{target}}$; see, e.g., Chung et al. (2022a;b); Song et al. (2022; 2023); Boys et al. (2023); Zhang et al. (2024). Using the decomposition of $p_{\text{target}}$, they draw approximate samples from $p_{\text{target}}$ based on approximations of the likelihood score (i.e., the difference of the score for the noised posterior and prior distributions) during inference. For instance, the common *reconstruction guidance* approximates this score by the (scaled) gradient of the log-likelihood evaluated at the denoised sample obtained via Tweedie's formula and the pre-trained model. While such plug-and-play approaches can yield impressive results for high-dimensional distributions without additional training, they typically lack theoretical guarantees and typically suffer from instabilities and mode collapse.

At the cost of simulating multiple particles during the generative process, the bias originating from approximating the likelihood score can be eliminated (in the limit of infinitely many particles) by leveraging ideas from SMC, i.e., by computing importance weights and interleaving the generative process with resampling steps (Wu et al., 2024). Taking into account an additional training phase, one can also obtain theoretical guarantees by writing the likelihood score as a solution to an stochastic optimal control (SOC) problem (as in DDS, however, with the pre-trained diffusion model as a reference process; see Didi et al. (2023, Section 2.4) and also Venkatraman et al. (2024). The SOC problem can then be solved using, e.g., the log-variance divergence. While such posterior sampling

---

[5]While Jing et al. (2022) use the probability flow ODE to obtain likelihoods, one could alternatively obtain importance weights in path space; see the references on diffusion-based samplers above.

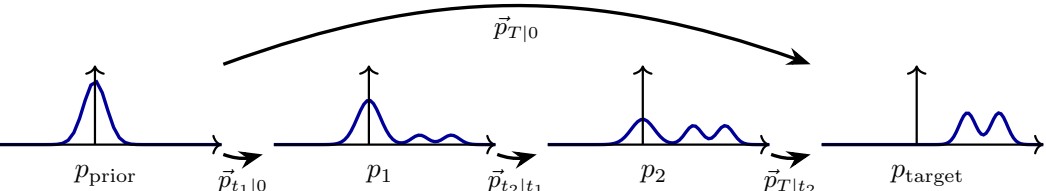

Figure 5: Illustration of annealed importance sampling along a geometric path, where we consider either one (top arrow) or three (bottom arrows) transition steps from the prior to the target.

approaches assume more structure than our considered sampling problem and rely on pre-trained diffusion prior, one could also adopt the idea of SCLD to such settings (see also Remark A.1), which we leave to future work. This would basically correspond to a combination of the approaches by Wu et al. (2024) and Didi et al. (2023), where the likelihood score is learned but training is facilitated by leveraging SMC steps.

We note that ideas similar to Didi et al. (2023) have recently also been used for fine-tuning diffusion models (where $p_{Y|X}(y|\cdot)$ corresponds to a reward function), using *adjoint matching* to minimize the KL divergence instead of, e.g., the log-variance divergence, to solve the SOC (Domingo-Enrich et al., 2024). While we propose to use the log-variance divergence to allow off-policy training and reduce variance (see Section 2.3), we note that adjoint matching and related approaches (Domingo-Enrich et al., 2023; Domingo-Enrich, 2024) could also be used for SCLD to solve the SOC problems in each subtrajectory.

## A.2 PROOFS AND THEORETICAL REMARKS

In this section, we provide additional remarks on our theory and the proof of Proposition 2.3.

**Remark A.1** (Generalizations)**.** We note that, in principle, prescribing an annealing, i.e. $p_{X^u} = \pi$, is not strictly necessary, and one could instead consider general bridges allowing for arbitrary density evolutions between the prior to the target. This, however, would come with the additional challenge of learning the (unnormalized) log-density $\log p_{X^u}$ of the controlled process, see, e.g., Richter & Berner (2024, Appendix A.7), which could make optimization potentially more difficult. Moreover, we can only use the approximate densities for the MCMC refinements as compared to using the target density $\pi$ in case of a prescribed annealing. While the general bridges do not exhibit unique solutions, one can consider the case studied in diffusion models, where the control $v$ of the reverse-time process in (7) is fixed such that $Y_0^v$ is approximately distributed as $p_{\text{prior}}$. Nelson's identity in (8) shows that it is sufficient to learn the log-density $\log p_{Y^v}$ and the optimal control can be computed using automatic differentiation, as leveraged in Phillips et al. (2024); Richter & Berner (2024). However, this can potentially be unstable and computationally more expensive.

**Remark A.2** (Connections to reinforcement learning)**.** The objectives of diffusion-based samplers can be viewed as stochastic optimal control problems; see, e.g., Dai Pra (1991); Zhang & Chen (2022); Berner et al. (2024). More generally, stochastic optimal control problems can be understood as versions of *maximum entropy reinforcement learning* in continuous time and space; see, e.g., Domingo-Enrich et al. (2024, Appendix C). Specifically, the prior distribution $p_{\text{prior}}$ together with the control $u$ define policies and transitions via the SDE (6) (or, in discrete time, via the transition kernels in (32) given by the Euler-Maruyama scheme). This allows the transfer of successful ideas from reinforcement learning to diffusion-based samplers. Motivated by previous work (Zhang et al., 2023a; Richter & Berner, 2024; Sendera et al., 2024), we propose to use *off-policy training* with *prioritized replay buffers* for SCLD, which is enabled by the log-variance loss (see Section 2.3).

*Proof of Proposition 2.3.* We follow the proof ideas from Nüsken & Richter (2021, Proposition 5.7), however, need to be careful since the reweighting of the measure $\vec{\mathbb{P}}_{[t_{n-1},t_n]} = \vec{\mathbb{P}}^{u,\pi_{n-1}}_{[t_{n-1},t_n]}$ is done w.r.t. a measure on the previous time interval $[t_m, t_{n-1}]$. Let us recall the KL divergence (16), namely

$$D := D_{\mathrm{KL}}\left(\vec{\mathbb{P}}_{[t_{n-1},t_n]}|\overset{\leftarrow}{\mathbb{P}}_{[t_{n-1},t_n]}\right) = -\mathbb{E}_{X\sim\vec{\mathbb{P}}_{[t_{n-1},t_n]}}\left[\log\left(w_{[t_{n-1},t_n]}(X)\right)\right]$$

$$= -\mathbb{E}_{X\sim\vec{\mathbb{P}}_{[t_m,t_n]}}\left[\log\left(w_{[t_{n-1},t_n]}(X)\right)w_{[t_m,t_{n-1}]}(X)\right],$$

where we abbreviate $w_{[s,t]} := \frac{\mathrm{d}\overset{\leftarrow}{\mathbb{P}}^{u,\pi(\cdot,s)}_{[s,t]}}{\mathrm{d}\vec{\mathbb{P}}^{u,\pi(\cdot,t)}_{[s,t]}}$. Using the analogous abbreviation $w^{\otimes I}_{[s,t]}$ for the product measures, we note that

$$\mathrm{Var}\left[\widehat{D}^{(K)}_{\mathrm{KL}}(\vec{\mathbb{P}}^{\otimes I}_{[t_{n-1},t_n]}|\overset{\leftarrow}{\mathbb{P}}^{\otimes I}_{[t_{n-1},t_n]})\right] = \frac{1}{K}\mathrm{Var}_{X\sim\vec{\mathbb{P}}^{\otimes I}_{[t_m,t_n]}}\left[\log\left(w^{\otimes I}_{[t_{n-1},t_n]}(X)\right)w^{\otimes I}_{[t_m,t_{n-1}]}(X)\right]$$

$$= \frac{M_I - D_I^2}{K}, \tag{21}$$

where

$$M_I := \mathbb{E}_{X\sim\vec{\mathbb{P}}^{\otimes I}_{[t_m,t_n]}}\left[\log^2\left(w^{\otimes I}_{[t_{n-1},t_n]}(X)\right)\left(w^{\otimes I}_{[t_m,t_{n-1}]}(X)\right)^2\right]$$

and

$$D_I := -\mathbb{E}_{X\sim\vec{\mathbb{P}}^{\otimes I}_{[t_m,t_n]}}\left[\log\left(w^{\otimes I}_{[t_{n-1},t_n]}(X)\right)w^{\otimes I}_{[t_m,t_{n-1}]}(X)\right]$$

$$= -\mathbb{E}_{X\sim\vec{\mathbb{P}}^{\otimes I}_{[t_{n-1},t_n]}}\left[\log\left(w^{\otimes I}_{[t_{n-1},t_n]}(X)\right)\right] = D_{\mathrm{KL}}\left(\vec{\mathbb{P}}^{\otimes I}_{[t_{n-1},t_n]}|\overset{\leftarrow}{\mathbb{P}}^{\otimes I}_{[t_{n-1},t_n]}\right) = ID. \tag{22}$$

Moreover, we can compute

$$M_I = \mathbb{E}_{X\sim\vec{\mathbb{P}}^{\otimes I}_{[t_m,t_n]}}\left[\left(\sum_{i=1}^{I}\log w^{(i)}_{[t_{n-1},t_n]}(X)\right)^2\left(w^{\otimes I}_{[t_m,t_{n-1}]}(X)\right)^2\right]$$

$$= \sum_{i=1}^{I}\mathbb{E}_{X\sim\vec{\mathbb{P}}^{\otimes I}_{[t_m,t_n]}}\left[\log^2\left(w^{(i)}_{[t_{n-1},t_n]}(X)\right)\left(w^{\otimes I}_{[t_m,t_{n-1}]}(X)\right)^2\right] \tag{23}$$

$$+ \sum_{\substack{i,j=1 \\ i\neq j}}^{I}\mathbb{E}_{X\sim\vec{\mathbb{P}}^{\otimes I}_{[t_m,t_n]}}\left[\log\left(w^{(i)}_{[t_{n-1},t_n]}(X)\right)\log\left(w^{(j)}_{[t_{n-1},t_n]}(X)\right)\left(w^{\otimes I}_{[t_m,t_{n-1}]}(X)\right)^2\right]$$

$$= IMC^{I-1} + I(I-1)D^2C^{I-2},$$

where $w^{(i)}_{[s,t]}$ denotes the weight for the $i$-th factor of the product measure and we abbreviate

$$M := \mathbb{E}_{X\sim\vec{\mathbb{P}}_{[t_m,t_n]}}\left[\log^2\left(w_{[t_{n-1},t_n]}(X)\right)\left(w_{[t_m,t_{n-1}]}(X)\right)^2\right] \geq D^2 \tag{24}$$

and

$$C := \mathbb{E}_{X\sim\vec{\mathbb{P}}_{[t_m,t_n]}}\left[\left(w_{[t_m,t_{n-1}]}(X)\right)^2\right] = \mathbb{E}_{X\sim\vec{\mathbb{P}}_{[t_m,t_{n-1}]}}\left[\left(w_{[t_m,t_{n-1}]}(X)\right)^2\right]$$

$$= D_{\chi^2}\left(\overset{\leftarrow}{\mathbb{P}}_{[t_m,t_{n-1}]}|\vec{\mathbb{P}}_{[t_m,t_{n-1}]}\right) + 1 \geq 1. \tag{25}$$

Combining the definition of the relative error with (21), (22), and (23), we obtain that

$$r^{(K)}\left(\vec{\mathbb{P}}^{\otimes I}_{[t_{n-1},t_n]}|\overset{\leftarrow}{\mathbb{P}}^{\otimes I}_{[t_{n-1},t_n]}\right) = \sqrt{\frac{M_I - D_I^2}{KD_I^2}} = \frac{C^{I/2}}{\sqrt{K}}\sqrt{\frac{MC + D^2(I-1)}{C^2ID^2} - \frac{1}{C^I}},$$

which, in view of (24) and (25), proves the claim. □

As already stated in the main text, we note that the log-variance divergence, defined in (18), does not scale exponentially in the dimension, as already proved in (Nüsken & Richter, 2021, Proposition 5.7). For convenience of the reader, let us explicitly verify that this statement also holds in our

setting. To this end, first note that

$$D_{\mathrm{LV}}^{\mathbb{Q}^{\otimes I}}\left(\vec{\mathbb{P}}_{[t_{n-1},t_n]}^{\otimes I}|\reflectbox{$\vec{\mathbb{P}}$}_{[t_{n-1},t_n]}^{\otimes I}\right) = \mathrm{Var}_{X\sim\mathbb{Q}^{\otimes I}}\left[\log\left(w_{[t_{n-1},t_n]}^{\otimes I}(X)\right)\right] = \tag{26a}$$

$$\sum_{i=1}^{I}\mathrm{Var}_{X\sim\mathbb{Q}}\left[\log\left(w_{[t_{n-1},t_n]}^{(i)}(X)\right)\right] = I D_{\mathrm{LV}}^{\mathbb{Q}}\left(\vec{\mathbb{P}}_{[t_{n-1},t_n]}|\reflectbox{$\vec{\mathbb{P}}$}_{[t_{n-1},t_n]}\right), \tag{26b}$$

where we recall that $\mathbb{Q}$ is an arbitrary reference measure. Following Cho et al. (2005), the sample variance satisfies

$$\mathrm{Var}\left[\widehat{D}_{\mathrm{LV}}^{\mathbb{Q}^{\otimes I},(K)}\left(\vec{\mathbb{P}}_{[t_{n-1},t_n]}^{\otimes I}|\reflectbox{$\vec{\mathbb{P}}$}_{[t_{n-1},t_n]}^{\otimes I}\right)\right] = \frac{1}{K}\left(\mu_4 - \frac{K-3}{K-1}D_{\mathrm{LV}}^{\mathbb{Q}^{\otimes I}}\left(\vec{\mathbb{P}}_{[t_{n-1},t_n]}^{\otimes I}|\reflectbox{$\vec{\mathbb{P}}$}_{[t_{n-1},t_n]}^{\otimes I}\right)^2\right), \tag{27}$$

where

$$\mu_4 = \mathbb{E}_{X\sim\mathbb{Q}^{\otimes I}}\left[\left(\log\left(w_{[t_{n-1},t_n]}^{\otimes I}(X)\right) - \mathbb{E}_{X\sim\mathbb{Q}^{\otimes I}}\left[\log\left(w_{[t_{n-1},t_n]}^{\otimes I}(X)\right)\right]\right)^4\right]. \tag{28}$$

We can calculate

$$\mu_4 = \mathbb{E}_{X\sim\mathbb{Q}^{\otimes I}}\left[\left(\sum_{i=1}^{I}\left(\log\left(w_{[t_{n-1},t_n]}^{(i)}(X)\right) - \mathbb{E}_{X\sim\mathbb{Q}}\left[\log\left(w_{[t_{n-1},t_n]}^{(i)}(X)\right)\right]\right)\right)^4\right] \tag{29a}$$

$$= I\,\mathbb{E}_{X\sim\mathbb{Q}}\left[\left(\log\left(w_{[t_{n-1},t_n]}(X)\right) - \mathbb{E}_{X\sim\mathbb{Q}}\left[\log\left(w_{[t_{n-1},t_n]}(X)\right)\right]\right)^4\right] \tag{29b}$$

$$+ 6\binom{I}{2}\mathbb{E}_{X\sim\mathbb{Q}}\left[\left(\log\left(w_{[t_{n-1},t_n]}(X)\right) - \mathbb{E}_{X\sim\mathbb{Q}}\left[\log\left(w_{[t_{n-1},t_n]}(X)\right)\right]\right)^2\right]^2, \tag{29c}$$

where we have used the fact that, for instance,

$$\mathbb{E}_{X\sim\mathbb{Q}^{\otimes I}}\left[\left(\log\left(w_{[t_{n-1},t_n]}^{(i)}(X)\right) - \mathbb{E}_{X\sim\mathbb{Q}}\left[\log\left(w_{[t_{n-1},t_n]}^{(i)}(X)\right)\right]\right)\right.$$
$$\left.\left(\log\left(w_{[t_{n-1},t_n]}^{(j)}(X)\right) - \mathbb{E}_{X\sim\mathbb{Q}}\left[\log\left(w_{[t_{n-1},t_n]}^{(j)}(X)\right)\right]\right)^3\right] = 0, \tag{30}$$

for $i \neq j$. Combining this with (26), it follows that $\mathrm{Var}\left[\widehat{D}_{\mathrm{LV}}^{(K)}\left(\vec{\mathbb{P}}_{[t_{n-1},t_n]}^{\otimes I}|\reflectbox{$\vec{\mathbb{P}}$}_{[t_{n-1},t_n]}^{\otimes I}\right)\right] = \mathcal{O}(I^2)$. Recalling the definition of the relative error, $r^{(K)} := \mathrm{Var}(\widehat{D}_{\mathrm{LV}}^{(K)})^{1/2}/D_{\mathrm{LV}}$, we see that it does not scale exponentially in $I$.

### A.3 ALGORITHMIC DETAILS AND PSEUDOCODE

We first provide formulas to compute the Radon-Nikodym derivative (RND) and the forward and backward kernels in discrete time. Then, we give an implementable method in Algorithm 2 and provide details on the resampling step and training with a buffer. Note that we can also use non-uniform discretizations within subtrajectories by adapting the times $hi$, $i = 0, \ldots, N$, accordingly.

#### A.3.1 COMPUTATION OF THE RADON-NIKODYM DERIVATIVE

As in Vargas et al. (2024), we obtain an approximate, computable formula for the Radon-Nikodym derivative in Lemma 2.2 between the $(n-1)$-th and $n$-th time step, given by

$$w_{[t_{n-1},t_n]}(X) = \frac{d\reflectbox{$\vec{\mathbb{P}}$}_{[t_{n-1},t_n]}}{d\vec{\mathbb{P}}_{[t_{n-1},t_n]}}(X) \approx \frac{\pi(X_{t_n},t_n)}{\pi(X_{t_{n-1}},t_{n-1})}\prod_{i=(n-1)L+1}^{nL}\frac{\reflectbox{$\vec{p}$}_{(i-1)h|ih}(X_{(i-1)h}|X_{ih})}{\vec{p}_{ih|(i-1)h}(X_{ih}|X_{(i-1)h})}, \tag{31}$$

where the transition densities for the forward and reverse-time SDEs, coming from the Euler-Maruyama discretization as in (19), are given as

$$\vec{p}_{t|s}(X_t|X_s) = \mathcal{N}(X_t; X_s + u(X_s,s)(t-s), \sigma^2(s)(t-s))$$
$$\reflectbox{$\vec{p}$}_{s|t}(X_s|X_t) = \mathcal{N}(X_s; X_t + (\sigma^2\nabla\log\pi - u)(X_t,t)(t-s), \sigma^2(t)(t-s)). \tag{32}$$

In practice, in line with Vargas et al. (2024), we parameterize the control as

$$u = \sigma^2\widetilde{u}_\theta + \frac{\sigma^2}{2}\nabla\log\pi, \tag{33}$$

where $\widetilde{u}_\theta$ is parametrized by a neural network. When $\widetilde{u}_\theta$ is initialized as the zero function, we recover an annealed form of Langevin dynamics (Welling & Teh, 2011), providing an improved starting point for optimization.

### A.3.2 A PRACTICAL ALGORITHM

In Algorithm 2, we give a practical and detailed version of Algorithm 1.

---

**Algorithm 2** SCLD-ForwardPass

---

**Require:** Target $\rho_{\text{target}}$, (learnable) prior $p_{\text{prior}} = \mathcal{N}(\mu_\theta, \text{diag}(\exp(2\ell_\theta)))$, number of subtrajectories $N$, steps
    per subtrajectory $L$ and step size $h$, annealing schedule $\beta_\theta$ as in (35), noise schedule $\sigma$, control $u$ given by
    neural network $\widetilde{u}^\theta$ as in (33), number of particles $K$

1: *Sample from prior (by reparametrization):* $\widehat{X}_0^{(1:K)} \sim p_{\text{prior}}$           ▷ Independent for each particle

2: *Initialize (unnormalized) importance weights:* $w_0^{(1:K)} = 1$

3: *Evaluate control and prior:* $u(\widehat{X}_0^{(1:K)}, 0)$ and $p_{\text{prior}}(\widehat{X}_0^{(1:K)})$

4: **for** $n = 1$ to $N$ **do**                                                ▷ Note that $t_n = nLh$

5:     **for** $i = (n-1)L + 1$ to $nL$ **do**                ▷ Consider the time interval $[(i-1)h, ih]$

6:         *Euler-Maruyama simulation:* $\widehat{X}_i^{(1:K)} \sim \vec{p}_{ih|(i-1)h}(\cdot | \widehat{X}_{i-1}^{(1:K)})$ as in (32)       ▷ See (19)

7:         *Evaluate control:* $u(\widehat{X}_i^{(1:K)}, ih)$

8:     *Evaluate (unnormalized) annealing:* $\pi(\widehat{X}_{nL}^{(1:K)}, t_n) = (p_{\text{prior}}^{1-\beta_\theta(t_n)} \rho_{\text{target}}^{\beta_\theta(t_n)})(\widehat{X}_{nL}^{(1:K)})$     ▷ See (20)

9:     *Compute RNDs:* $w_{[t_{n-1}, t_n]}^{(1:K)}$ as in (31)                 ▷ For every $k$, we use $X_{ih} = \widehat{X}_i^{(k)}$

10:     *Update weights:* $w_n^{(1:K)} = w_{n-1}^{(1:K)} w_{[t_{n-1}, t_n]}^{(1:K)}$

11:     *Resample:* $\widehat{X}_{nL}^{(1:K)}, w_n^{(1:K)} = \texttt{resample}(\widehat{X}_{nL}^{(1:K)}, w_n^{(1:K)})$            ▷ See Algorithm 5

12:     *MCMC step:* Update $\widehat{X}_{nL}^{(1:K)}$ with $\pi(\cdot, t_n)$-invariant kernel

13: **return** RNDs $(w_{[t_{n-1}, t_n]}^{(1:K)})_{n=1}^N$, weights $(w_n^{(1:K)})_{n=0}^N$, trajectories $\widehat{X}^{(1:K)}$,
        $\log Z$ estimate $\sum_{n=1}^N \log\left(\sum_{k=1}^K w_{n-1}^{(k)} w_{[t_{n-1}, t_n]}^{(k)}\right)$, ELBO $\sum_{n=1}^N \sum_{k=1}^K w_{n-1}^{(k)} \log\left(w_{[t_{n-1}, t_n]}^{(k)}\right)$

---

**Prioritized replay buffer.** We give the exact algorithm of our replay buffer in Algorithm 4. We note that there are many alternative possibilities for choosing the buffer priority (including by importance weight), which we leave to future exploration. Moreover, as in traditional replay buffers (Mnih, 2013), there is an option to perform multiple gradient steps per simulation to reduce computation costs.

**Resampling.** The work of Webber (2019) shows that there is great scope to design resampling methods. However, in line with prior work, we opt to use the simple adaptive multinomial resampling for which pseudocode is provided in Algorithm 5.

**Annealing Schedule** While we choose to learn the annealing schedule, an alternative approach that takes advantage of the flexibility of our continuous-time perspective is to set $\beta(t) = t$ and instead adaptively choose the time discretization. For example, the constant-ESS scheme annealing schedule Buchholz et al. (2021) is a commonly used scheme in the SMC literature and can be easily adapted to the SCLD setting for both training and sampling. We leave the investigation of this idea to future work.

### A.4 BENCHMARK TARGET DISTRIBUTIONS

Here, we introduce the target densities considered in our experiments more formally. Most of these are standard benchmarks taken from, e.g., Heng et al. (2017); Arbel et al. (2021); Geffner & Domke (2022); Richter & Berner (2024); Blessing et al. (2024).

### A.4.1 BAYESIAN STATISTICS TASKS

For these tasks, no groundtruth samples are available.

---

**Algorithm 3** SCLD-Training $\qquad\qquad\qquad\qquad$ ▷ See Algorithm 4 for training with buffers.

---

**Require:** Number of iterations $I$, initial parameters $\theta^{(0)}$, optimizer update update, inputs for Algorithm 2

1: **for** $i = 0$ to $I - 1$ **do**
2: $\qquad$ *Run Algorithm 2:* $(w^{(1:K)}_{[t_{n-1},t_n]})^N_{n=1}, (w^{(1:K)}_n)^N_{n=0} = \text{SCLD-ForwardPass}(\theta^{(i)})$
3: $\qquad$ **if** LV **then** $\qquad\qquad\qquad\qquad\qquad$ ▷ Trajectories $\widehat{X}^{(1:K)}$ are detached during forward pass
4: $\qquad\qquad$ *Compute loss:* $\mathcal{L} = \sum^N_{n=1} \frac{1}{K} \sum^K_{k=1} \left( \log w^{(k)}_{[t_{n-1},t_n]} - \frac{1}{K} \sum^K_{i=1} \log w^{(i)}_{[t_{n-1},t_n]} \right)^2$
5: $\qquad$ **else if** KL **then**
6: $\qquad\qquad$ *Compute loss:* $\mathcal{L} = -\sum^N_{n=1} \frac{1}{K} \sum^K_{k=1} \text{detach}(w^{(k)}_{n-1}) \log w^{(k)}_{[t_{n-1},t_n]}$
7: $\qquad$ *Compute gradient w.r.t. parameters:* $G^{(i)} = \nabla_{\theta^{(i)}} \mathcal{L}$
8: $\qquad$ *Optimizer step:* $\theta^{(i+1)} = \text{update}(\theta^{(i)}, (G^{(j)})^i_{j=0})$ $\qquad\qquad\qquad\qquad$ ▷ We use Adam
9: **return** Optimized parameters $\theta^{(I)}$

---

**Algorithm 4** SCLD-Buffer-Training

---

**Require:** Buffer $(\mathcal{B}_n)^N_{n=1}$ for every sutrajectory, inputs for Algorithm 3

1: **for** $i = 0$ to $I - 1$ **do**
2: $\qquad$ *Run Algorithm 2:* $(w^{(1:K)}_{[t_{n-1},t_n]})^N_{n=1}, \widehat{X}^{(1:K)} = \text{SCLD-ForwardPass}(\theta^{(i)})$ with $K$ particles
3: $\qquad$ **for** $n = 1$ to $N$ **do**
4: $\qquad\qquad$ *Store subtrajectories:* $(\widehat{X}^{(1:K)}_i)^{nL}_{i=(n-1)L}$ into $\mathcal{B}_n$ with weights $w^{(1:K)}_{[t_{n-1},t_n]}$, replacing oldest entries
5: $\qquad\qquad$ *Sample from buffer:* $\widetilde{X}^{(1:K/2)} \sim \mathcal{B}_n$ with probability proportional to buffer weights
6: $\qquad\qquad$ *Recompute RNDs:* $\widetilde{w}^{(1:K/2)}_{[t_{n-1},t_n]}$ for detached $\widetilde{X}^{(1:K/2)}$ using (31) and current parameters $\theta^{(i)}$
7: $\qquad\qquad$ *Update buffer:* Set $\widetilde{w}^{(1:K/2)}_{[t_{n-1},t_n]}$ as weights for $\widetilde{X}^{(1:K/2)}$ $\qquad\qquad$ ▷ Updating all $B$ particles is too slow
8: $\qquad\qquad$ *Sample other half from simulation:* $\widetilde{w}^{(K/2+1:K)}_{[t_{n-1},t_n]}$ from $w^{(1:K)}_{[t_{n-1},t_n]}$ uniformly without replacement
9: $\qquad$ *Compute log-variance loss:* $\mathcal{L} = \sum^N_{n=1} \frac{1}{K} \sum^K_{k=1} \left( \log \widetilde{w}^{(k)}_{[t_{n-1},t_n]} - \frac{1}{K} \sum^K_{i=1} \log \widetilde{w}^{(i)}_{[t_{n-1},t_n]} \right)^2$
10: $\qquad$ *Compute gradient w.r.t. parameters:* $G^{(i)} = \nabla_{\theta^{(i)}} \mathcal{L}$
11: $\qquad$ *Optimizer step:* $\theta^{(i+1)} = \text{update}(\theta^{(i)}, (G^{(j)})^i_{j=0})$ $\qquad\qquad\qquad\qquad$ ▷ We use Adam
12: **return** Optimized parameters $\theta^{(I)}$

---

**Bayesian Logistic Regression (Sonar and Credit).** We used two binary classification problems in our benchmark, which have also been used in various other works to compare different state-of-the-art methods in variational inference and MCMC. Specifically, we assess the performance of a Bayesian logistic model with

$$\rho_{\text{target}}(x) = p(x) \prod_{i=1}^n \text{Bernoulli}(y_i; \text{sigmoid}(x \cdot u_i))$$

on two standardized datasets $((u_i, y_i))^n_{i=1}$, namely Sonar ($d = 61$) and German Credit ($d = 25$) with $n = 208$ and $n = 1000$ data points, respectively. We choose $p = \mathcal{N}(0, I)$ for Sonar and $p \equiv 1$ for Credit (in line with the code of Blessing et al. (2024) which omitted the prior).

**Random Effect Regression (Seeds).** The Seeds ($d = 26$) target uses a random effect regression model given by:

$$\tau \sim \text{Gamma}(0.01, 0.01)$$
$$a_0, a_1, a_2, a_{12} \sim \mathcal{N}(0, 10)$$
$$b_i \sim \mathcal{N}\left(0, \frac{1}{\sqrt{\tau}}\right), \quad i = 1, \dots, 21,$$
$$\text{logits}_i = a_0 + a_1 x_i + a_2 y_i + a_{12} x_i y_i + b_1, \quad i = 1, \dots, 21,$$
$$r_i \sim \text{Binomial}(\text{logits}_i, N_i), \quad i = 1, \dots, 21.$$

The goal is to do inference over the variables $\tau, a_0, a_1, a_2, a_{12}$ and $b_i$ for $i = 1, \dots, 21$, given observed values for $x_i, y_i,$ and $N_i$ from a dataset modeling the germination proportion of seeds; see Geffner & Domke (2022) for details.

---

**Algorithm 5** Adaptive Multinomial Resampling

---

**Require:** particles $X^{(1:K)}$, unnormalized weights $w^{(1:K)}$
1: *Normalize:* $W^{(k)} = w^{(k)} / \sum_{i=1}^{K} w^{(i)}$, $k = 1, \ldots, K$
2: *Compute ESS:* $\text{ESS} = 1 / \sum_{k=1}^{K} (W^{(k)})^2$
3: **if** $\text{ESS} < \alpha K$ **then**                                               ▷ We take $\alpha = 0.3$
4:     **for** $k = 1$ to $K$ **do**
5:         *Sample index from categorical distribution:* $i \in \{1, \ldots, K\}$ with probabilities $W^{(1:K)}$
6:         *Define resampled particle:* $\widetilde{X}^{(k)} = X^{(i)}$
7:     *Reset weights:* $W^{(1:K)} = 1/K$
8: **else**
9:     *Keep particles:* $\widetilde{X}^{(1:K)} = X^{(1:K)}$
10: **return** resampled particles $\widetilde{X}^{(1:K)}$, updated and normalized weights $W^{(1:K)}$

---

**Time Series Models (Brownian).** The Brownian ($d = 32$) model corresponds to the time discretization of a Brownian motion with Gaussian observation noise:

$$\alpha_{\text{inn}} \sim \text{LogNormal}(0, 2),$$
$$\alpha_{\text{obs}} \sim \text{LogNormal}(0, 2),$$
$$x_1 \sim \mathcal{N}(0, \alpha_{\text{inn}}),$$
$$x_i \sim \mathcal{N}(x_{i-1}, \alpha_{\text{inn}}), \quad i = 2, \ldots, 30,$$
$$y_i \sim \mathcal{N}(x_i, \alpha_{\text{obs}}), \quad i = 1, \ldots, 30.$$

Inference is performed over the variables $\alpha_{\text{inn}}$, $\alpha_{\text{obs}}$, and $\{x_i\}_{i=1}^{30}$ given the observations $\{y_i\}_{i=1}^{10}$ and $\{y_i\}_{i=20}^{30}$ (i.e., the middle observations are missing); see Geffner & Domke (2022).

**Spatial Statistics (LGCP).** The *Log Gaussian Cox process* (LGCP) is a popular high-dimensional task in spatial statistics (Møller et al., 1998), which models the position of pine saplings. Using a $d = 40 \times 40 = 1600$ grid, we obtain the unnormalized target density by

$$\rho_{\text{target}} = \mathcal{N}(x; \mu, \Sigma) \prod_{i=1}^{d} \exp \left( x_i y_i - \frac{\exp (x_i)}{d} \right),$$

where $y$ is a given dataset and $\mu$ and $\Sigma$ are the mean and covariance matrix of the given prior. We use the more challenging unwhitened version; see Heng et al. (2017); Arbel et al. (2021) for details.

### A.4.2 SYNTHETIC TARGETS

For these tasks, groundtruth samples are available.

**Robot.** The Robot targets (Arenz et al., 2020) (Robot1, Robot4) aim at learning joint configurations of a 10 degrees-of-freedom planar robot, parameterized by

$$\alpha = (\alpha_1, \ldots, \alpha_{10}),$$

such that it reaches a desired goal position while enforcing smooth configurations. The target density is given by

$$\rho_{\text{target}}(\alpha) = p_{\text{conf}}(\alpha) p_{\text{cart}}(\alpha),$$

where $p_{\text{conf}}$ enforces smooth configurations and $p_{\text{cart}}$ penalizes deviations from the goal position. $p_{\text{conf}}$ is modeled as zero-mean Gaussian distribution with a diagonal covariance matrix, where the angle $\alpha_1$ of the first joint has a variance of 1 and the remaining joint angles $\alpha_2, \ldots, \alpha_{10}$ have a variance of $4 \times 10^{-2}$.

Formally, we define the locations of the robot joints by

$$x_i(\alpha) = \sum_{j=1}^{i} \cos(\alpha_j), \qquad i = 0, \ldots, 10,$$

$$y_i(\alpha) = \sum_{j=1}^{i} \sin(\alpha_j), \qquad i = 0, \ldots, 10.$$

In the Robot1 task there is one goal at $(7, 0)$, and we specify

$$p_{\text{cart}}(\alpha) = \mathcal{N}\left(\begin{pmatrix} x_{10}(\alpha) \\ y_{10}(\alpha) \end{pmatrix}; \begin{pmatrix} 7 \\ 0 \end{pmatrix}, 10^{-4} I\right), \tag{34}$$

i.e., a Gaussian distribution centered at the Cartesian coordinates of the goal position, with a variance of $10^{-4}$ in both directions.

In the Robot4 task there are 4 goals at $(\pm 7, 0)$ and $(0, \pm 7)$, and so $p_{\text{cart}}$ is given by the maximum over the four respective Gaussian distributions as in (34) (up to a constant of proportionality). Groundtruth samples are generated by long *slice sampling* runs (Neal, 2003) and taken from the repository of Arenz et al. (2020).

**Mixture distributions (GMM and MoS).** For the GMM and MoS tasks, we define a mixture distribution with $m$ components as

$$p_{\text{target}} = \frac{1}{m} \sum_{i=1}^{m} p_i.$$

The Gaussian Mixture Model (GMM), taken from Blessing et al. (2024), consists of $m = 40$ mixture components with

$$p_i = \mathcal{N}(\mu_i, I),$$
$$\mu_i \sim \mathcal{U}_d(-40, 40),$$

where $\mathcal{U}_d(l, u)$ refers to a uniform distribution on $[l, u]^d$. We take $d = 50$ for the main experiments.

The Mixture of Student's t-distributions (MoS), taken from Blessing et al. (2024), comprises $m = 10$ Student's t-distributions $t_2$, where the 2 refers to the degree of freedom. Specifically, we use

$$p_i = t_2 + \mu_i,$$
$$\mu_i \sim \mathcal{U}_d(-10, 10),$$

where $\mu_i$ refers to the translation of the individual components, and take $d = 50$. For both the GMM and MoS tasks, the $\mu_i$'s are fixed throughout experiments, i.e., selected with the same random seed.

**Funnel.** The Funnel target introduced in Neal (2003) is a challenging funnel-shaped distribution given by

$$p_{\text{target}}(x) = \mathcal{N}(x_1; 0, \sigma^2) \mathcal{N}(x_2, \ldots, x_{10}; 0, \exp(x_1) I),$$

with $\sigma^2 = 9$ for any number of dimensions $d \geq 2$. We take $d = 10$ in our main experiments.

**Many-Well (MW).** A typical problem in molecular dynamics considers sampling from the stationary distribution of Langevin dynamics. In our example we shall consider a $d$-dimensional many-well potential, corresponding to the (unnormalized) density

$$\rho_{\text{target}}(x) = \exp\left(-\sum_{i=1}^{m}(x_i^2 - \delta)^2 - \frac{1}{2}\sum_{i=m+1}^{d} x_i^2\right).$$

In line with Berner et al. (2024); Sun et al. (2024), we take $d = 5$, $m = 5$, and $\delta = 4$, leading to $2^m = 32$ well-separated modes. Groundtruth $\log Z$ and samples can be obtained by noting that the distribution factors over dimensions.

## A.5 EXPERIMENTAL DETAILS

In this section, we describe the experimental setup and evaluation protocol. We also discuss design choices for our main experiments as well as how our hyperparameters are selected.

### A.5.1 METRICS AND EVALUATION

- **Maximization of the ELBO.** The ELBO refers to a lower bound on $\log Z$. This is a classic benchmark for samplers, and higher ELBOs are usually associated with precise sampling from discovered modes. However, the ELBO is not necessarily indicative of mode collapse; see Blessing et al. (2024) and Appendix A.6.4 for details.

- **Minimization of the Sinkhorn distance.** The Sinkhorn distance $\mathcal{W}_2$ is an optimal transport (OT) distance. When computed between a set of generated samples and a groundtruth set of samples

from the target (when the latter is available), this gives an estimate of the OT distance from the distribution generated by the sampler to $p_{\text{target}}$. As discussed further in Blessing et al. (2024), low OT distances are associated with good mode coverage (i.e., avoiding mode collapse).

For both ELBO and optimal transport evaluation, we follow the protocol of Blessing et al. (2024). In particular, we use the Sinkhorn distance as implemented in Cuturi et al. (2022) and use standard formulas for the ELBO computations of our baselines. For SCLD, the ELBO computation is stated in Algorithm 2. We compute all performance criteria 100 times during training using 2000 samples, applying a running average with a length of 5 over these evaluations to obtain robust results within a single run. To ensure robustness across runs, we use four different random seeds and average the best results from each run. As we use the same evaluation protocol as Blessing et al. (2024), we re-use their results for DDS and PIS whenever available.

As discussed in Blessing et al. (2024), ELBO metrics are insensitive to mode collapse and, as such, may not accurately reflect the quality of samples on multimodal tasks. As groundtruth samples are available for the synthetic tasks considered and due to their generally multimodal nature, we report Sinkhorn distances for these tasks.

### A.5.2 DESIGN CHOICES

We follow the following principles:

- SCLD follows design choices of other methods when these are shared.

- SCLD reuses the hyperparameter choices of baseline methods when shared such that it is not tuned excessively.

- Baseline methods should be given as much or more computational budget compared to SCLD.

**General remarks.** For SCLD, CMCD, DDS, and PIS we take the convention that $T = 1$ as rescaling time is equivalent to rescaling the noise level. Since the objectives of DDS and the *Time-Reversed Diffusion Sampler* (DIS) (Berner et al., 2024) only differ by choice of the reference process (see also Berner et al. (2024, Appendix A.10.1), Richter & Berner (2024, Section 3), and Vargas et al. (2024, Appendix C.3)), we do not explicitly compare against DIS in this work.

**CMCD and SCLD.** As SCLD and CMCD share numerous design choices, we mostly follow the choices of CMCD as in Vargas et al. (2024). In particular, we opt to learn the prior as well as the annealing schedule. For the former, we define $p_{\text{prior}} \coloneqq \mathcal{N}(\mu_\theta, \text{diag}(\exp(2\ell_\theta)))$. In other words, we parameterize the Gaussian prior through its mean $\mu_\theta \in \mathbb{R}^d$ and logarithmic standard deviations $\ell_\theta \in \mathbb{R}^d$, initialized to $\mathcal{N}(0, \sigma^2 I)$, i.e., $\mu_\theta = 0$ and $(\ell_\theta)_i = \log(\sigma)$, for some $\sigma > 0$ (referred to as initial scale) to be tuned. We update $\mu_\theta$ and $\ell_\theta$ via the parameterization trick as training progresses. For learning the annealing, we parameterize the schedule in (20) for every $j \in \{1, \ldots, NL\}$ by

$$\beta_\theta(jh) \coloneqq \sum_{i=1}^{j} \frac{\text{softplus}(\theta_i)}{\sum_{i=1}^{NL} \text{softplus}(\theta_i)}, \tag{35}$$

where $\theta_i \in \mathbb{R}$ are learnable parameters. We choose the buffer size to be 20 times the training batch size, i.e., $B = 20K$. Moreover, we parametrize the control $u$ as in (33). For SCLD, we use the subtrajectory settings from Section 3.

**CRAFT.** We use the implementation by Blessing et al. (2024), following the standard settings of Matthews et al. (2022). Specifically, we employ diagonal affine flows as the transport maps.

**SMC operations.** We use the same resampling strategy and MCMC kernel for CRAFT, SMC, and SCLD. In particular, every SMC step consists of adaptive resampling with a threshold of $0.3K$, followed by one Hamiltonian Monte Carlo (HMC) step with 10 leapfrog steps. For details on the advanced SMC schemes (SMC-ESS and SMC-FC), we refer to Buchholz et al. (2021) and Appendix A.6.8.

**Optimization and batch size.** We utilize the Adam optimizer for all methods that require learning. We also found that clipping gradients to 1 was important for stable training on all diffusion-based methods. We use batch size 2000 for training except for LGCP, where batch size 300 is used. We always evaluate with $K = 2000$ particles.

**Number of annealing / diffusion steps.** For SMC, DDS, PIS, CMCD, and SCLD in the main experiments, we fix 128 steps. In particular, we have $L = 128/N$ for SCLD. For CRAFT, we sweep over $[4, 8, 128]$ annealing steps (which also define the number of SMC operations).

**Number of training iterations.** We select the number of training iterations such that all methods are given roughly the same number of target function evaluations (NFEs) for a given number of SMC operations or subtrajectories $N$, evaluations per SMC operation $M$, and annealing or diffusion steps per subtrajectory $L$. In our setup, $M = 10$ due to the 10 leapfrog steps in HMC, $(N, L) = (1, 128)$ for DDS, PIS, and CMCD, and $N \in \{4, 8, 128\}$ for CRAFT (with $L = 1$) and SCLD (with $L = 128/N$). As a reference value, we use $40000$ iterations for DDS and PIS as in Blessing et al. (2024). We report the chosen number of iterations for each method in Table 4.

Table 4: Number of training iterations for our considered method depending on the number of SMC operations or subtrajectories $N$. The last rows show the approximate number of target function evaluations (NFEs) per particle in each iteration w.r.t. the number of evaluations per SMC operation $M$ and annealing or diffusion steps per subtrajectory $L$.

|  | CRAFT | SCLD | DDS, PIS, CMCD-KL, CMCD-LV |
|---|---|---|---|
| $N = 1$ | – | – | $4 \times 10^4$ |
| $N = 4$ | $10^5$ | $2.5 \times 10^4$ | – |
| $N = 8$ | $5 \times 10^4$ | – | – |
| $N = 128$ | $3 \times 10^3$ | $3 \times 10^3$ | – |
| Approx. NFEs per particle | MN | MN + L | L |
| Our setup | $L = 1, M = 10$ | $LN = 128, M = 10$ | $N = 1, L = 128$ |

We note that all baselines converged satisfactorily within the given iteration budget. Moreover, the generous budget of $40000$ iterations for DDS, PIS, and CMCD required running for $4 - 20$ times as long as SCLD's training process on equivalent architecture for our considered tasks (see also Table 9).

### A.5.3 HYPERPARAMETER SELECTION

**General remarks.** We follow the spirit of experimental design in Blessing et al. (2024) to fairly compare SCLD with our diverse range of baselines. We describe the search space and selection procedure below. We select the best configuration based on the target metric and a single seed. We note that alternative experimental setups such as done in Vargas et al. (2024) are possible, leveraging the ability of CMCD and SCLD to learn further hyperparameters end-to-end or use variational mean field approximations (see Appendix A.6.6) instead of a grid-search.

**Prior scale.** For all methods that require a $\mathcal{N}(0, \sigma^2 I)$ prior, we sweep over $\sigma$ in $[0.1, 1, 10]$ for tasks where we have no information about the target. For GMM40 and MoS tasks, we know that the initial scale should be around 40 and 15, respectively, by construction of the problem, so we fix these values for all methods. Similarly, for the Robot tasks, we know that the coordinates correspond to radial angles, so we set the initial scale to 2 to cover the $[-\pi, \pi]$ range.

**Diffusion noise schedule.** For diffusion-based samplers a noise schedule $\sigma$ as in (6) needs be specified. For PIS, we use a linear noise schedule as in Zhang & Chen (2022), and for DDS, CMCD, and SCLD we use a cosine schedule as in Vargas et al. (2023). Both noise schedules are parameterized by a "minimum diffusion" and a "maximum diffusion" coefficient. We set the minimum diffusion noise level to 0.01 for all tasks and methods except the Robot tasks, where we set it to 0.001. For all methods and tasks we perform grid searches over the maximum diffusion parameter. For all tasks except the Robot and GMM40 tasks we search in $[0.1, 1, 10]$. Due to the large initial scale of GMM40, we search the maximum diffusion parameter over $[5, 10, 20]$. For Robot, we search it in $[0.003, 0.03, 0, 3]$ instead of the usual grid due to the constructed sharpness of the modes.

**Architecture.** For hyperparameter selection on CMCD-KL, CMCD-LV and SCLD, we use the PIS-GradNet architecture (with detached score and 2 hidden layers of 64 units) for all diffusion-based methods as in Vargas et al. (2023). However, for CMCD-KL, we found that using the simpler MLP architecture described in Vargas et al. (2024) (which we term PISNet) gave significantly better performance than PISGradNet on most tasks. As such, to ensure strong baselines for CMCD-KL and CMCD-LV, we also select the best architecture among PISGradNet and PISNet (with 2 hidden layers

of 90 units to ensure similar parameter counts), re-sweeping learning rates as necessary. For SCLD we use PISGradNet on all tasks.

**CMCD-KL, CMCD-LV, DDS, and PIS.** We jointly grid search the initial scale and maximum diffusion along with the learning rates. We use one learning rate for the model $\widetilde{u}_\theta$ and prior $p_{\text{prior}}$, and another for the annealing schedule $\beta$. We sweep over the learning rate of the model in $[10^{-3}, 10^{-4}, 10^{-5}]$ and learning rate of the annealing schedule in $[10^{-2}, 10^{-3}]$. We perform model selection using 8000 gradient steps instead of 40000 due to the large grid.

**SMC.** For all tasks not present in Blessing et al. (2024) (namely the Robot tasks and MW54), we search the same parameter grid used for other methods for the scale of the prior, jointly with HMC step sizes. For all tasks present in the benchmark, we re-use their results and SMC configuration for SCLD and CRAFT. We tuned the step size of HMC, using different step sizes for $t < T/2$ and $t > T/2$ (where time corresponds to annealing steps in CRAFT) in the same fashion as Blessing et al. (2024). We search step sizes in the set $[0.001, 0.01, 0.5, 0.1, 0.2]$

**CRAFT.** We sweep over $[4, 8, 128]$ for the number of annealing steps, jointly with the prior scale and the learning rate (also in $[10^{-3}, 10^{-4}, 10^{-5}]$), and choose the best value. As in Blessing et al. (2024), we re-use the HMC step sizes that were tuned for SMC. Our results uniformly reproduce or improve upon those presented in the aforementioned paper due to the extended search space.

**SCLD.** To ensure a fair comparison with baseline methods, we reuse the chosen scale and diffusion parameters of CMCD-LV as well as the HMC step sizes tuned for SMC. The only grid search we perform for SCLD is over the learning rate of the model in $[10^{-3}, 10^{-4}]$ and the learning rate of the annealing schedule in $[10^{-2}, 10^{-3}]$. However, as reflected in Table 5, setting all learning rates to $10^{-3}$ typically turned out to be a robust choice.

**Table of hyperparameter choices.** In Tables 5 and 6 we present the tuned hyperparameters we obtained. Please note that "PGN" refers to the PISGradNet architecture, whereas "PN" refers to the PISNet architecture. We refer to Blessing et al. (2024) for further details and design choices for PIS and DDS. In Table 6, we specify the hyperparameters for DDS, PIS, and SMC on tasks not present in Blessing et al. (2024).

Table 5: Hyperparameter choices of our considered methods for the tasks in Blessing et al. (2024).

| | Funnel (10d) | MW54 (5d) | Robot1 (10d) | Robot4 (10d) | GMM40 (50d) | MoS (50d) | Seeds (26d) | Sonar (61d) | Credit (25d) | Brownian (32d) | LGCP (1600d) |
|---|---|---|---|---|---|---|---|---|---|---|---|
| **CMCD-KL** | | | | | | | | | | | |
| Initial Scale | 1.0 | 1.0 | 2.0 | 2.0 | 40.0 | 15.0 | 1.0 | 0.1 | 10.0 | 0.1 | 1.0 |
| Max Diffusion | 10.0 | 1.0 | 0.03 | 0.03 | 10.0 | 10.0 | 1.0 | 1.0 | 1.0 | 1.0 | 10.0 |
| Architecture | PN | PGN | PGN | PGN | PN | PGN | PN | PN | PGN | PN | PGN |
| Model LR | 0.001 | 0.0001 | 0.001 | 0.001 | 0.0001 | 0.001 | 0.001 | 0.001 | 0.001 | 0.001 | 0.0001 |
| Annealing Schedule LR | 0.01 | 0.001 | 0.001 | 0.001 | 0.01 | 0.001 | 0.01 | 0.001 | 0.01 | 0.01 | 0.01 |
| **CMCD-LV** | | | | | | | | | | | |
| Initial scale | 1.0 | 1.0 | 2.0 | 2.0 | 40.0 | 15.0 | 1.0 | 1.0 | 0.1 | 0.1 | 1.0 |
| Maximum diffusion | 1.0 | 10.0 | 0.03 | 0.03 | 20.0 | 1.0 | 1.0 | 1.0 | 0.1 | 1.0 | 10.0 |
| Architecture | PN | PN | PGN | PGN | PGN | PN | PGN | PN | PN | PGN | PGN |
| Model LR | 0.001 | 0.0001 | 0.0001 | 0.001 | 0.0001 | 0.001 | 0.001 | 0.001 | 0.001 | 0.001 | 0.0001 |
| Annealing schedule LR | 0.01 | 0.01 | 0.01 | 0.01 | 0.01 | 0.001 | 0.01 | 0.001 | 0.01 | 0.01 | 0.001 |
| **SCLD** | | | | | | | | | | | |
| Model LR | 0.001 | 0.0001 | 0.001 | 0.001 | 0.001 | 0.001 | 0.001 | 0.001 | 0.001 | 0.001 | 0.001 |
| Annealing schedule LR | 0.01 | 0.01 | 0.01 | 0.001 | 0.001 | 0.001 | 0.01 | 0.01 | 0.01 | 0.001 | 0.001 |
| **CRAFT** | | | | | | | | | | | |
| Number of steps | 128 | 128 | 8 | 4 | 4 | 128 | 128 | 128 | 8 | 128 | 128 |
| LR | 0.001 | 0.00001 | 0.001 | 0.001 | 0.00001 | 0.001 | 0.0001 | 0.0001 | 0.0001 | 0.001 | 0.001 |
| initial scale | 1.0 | 1.0 | 2.0 | 2.0 | 40.0 | 15.0 | 0.1 | 1.0 | 1.0 | 1.0 | 1.0 |

**Experimental details.** Here, we provide additional details on the experiments in the main part of the paper.

- **Improved convergence (Figure 3).** All experiments were performed on a single Nvidia RTX4090 GPU using the same settings as the main experiments.

Table 6: Hyperparameter choices of DDS, PIS, and SMC for the tasks not present in Blessing et al. (2024).

|  | Robot1 | Robot4 | MW54 |
|---|---|---|---|
| **SMC** | | | |
| Initial scale | 2.0 | 2.0 | 1.0 |
| HMC step sizes | [0.001, 0.01] | [0.01, 0.001] | [0.01, 0.001] |
| **DDS** | | | |
| Initial scale | 2.0 | 2.0 | 0.1 |
| Maximum diffusion | 0.3 | 0.3 | 10.0 |
| LR | 0.001 | 0.001 | 0.00001 |
| **PIS** | | | |
| Maximum diffusion | 0.3 | 0.3 | 10.0 |
| LR | 0.00001 | 0.00001 | 0.00001 |

- **Varying the number of SMC steps (Figure 4).** For this study, we train for 8000 gradient steps in all instances and vary the number of subtrajectories at training and evaluation time. Apart from that, we use the same hyperparameters and procedures as in the main experiments. In particular, the total number of annealing steps is fixed to 128.

## A.6 ADDITIONAL EXPERIMENTS

In this section, we present additional experiments.

### A.6.1 ABLATION STUDIES OF SCLD

In Figure 6, we study the effect of removing various parts of SCLD on several tasks. We investigate the use of the buffer, resampling, and MCMC steps. For this experiment, all other design choices are kept the same as in the main experiments. In particular, the reported results for the full SCLD algorithm here coincide with those in the main experiments up to variation due to seeds. On the other hand, the "No (Buffer,Resampling,MCMC)" Algorithm corresponds to CMCD-LV with subtrajectories.

In all studied cases except the Seeds task, the addition of each component (MCMC, resampling, and buffer) improves performance (we use a logarithmic scale for clarity on the Robot task). In the case of the Seeds task, the performances of all choices are effectively the same (note the small range of the y-axis). In summary, this study shows that none of our components are redundant.

### A.6.2 REMOVING MCMC COMPONENTS

Here, we investigate the effect of not using MCMC steps during training. This is an interesting question because, unlike SMC methods with deterministic transitions like CRAFT, where MCMC steps are needed to remove the particle degeneracy caused by resampling steps, our stochastic transitions do this automatically. As such, it is possible to remove MCMC steps from the SCLD training procedure, and we investigate the effect of doing so here, as it offers potentially accelerated training.

Table 7: ELBOs attained by SCLD when removing MCMC steps during training and evaluation.

| ELBO ($\uparrow$) | Seeds (26d) | Sonar (61d) | Credit (25d) | Brownian (32d) | LGCP (1600d) |
|---|---|---|---|---|---|
| **SCLD** | $-73.45_{\pm 0.01}$ | $-108.17_{\pm 0.25}$ | $-504.46_{\pm 0.09}$ | $1.00_{\pm 0.18}$ | $486.77_{\pm 0.70}$ |
| **SCLD-NoMCMC** | $-73.48_{\pm 0.03}$ | $-109.39_{\pm 1.10}$ | $-504.72_{\pm 0.34}$ | $0.82_{\pm 0.09}$ | $415.83_{\pm 19.53}$ |

Table 8: Sinkhorn distances attained by SCLD when removing MCMC steps during training and evaluation.

| Sinkhorn ($\downarrow$) | Funnel (10d) | MW54 (5d) | Robot1 (10d) | Robot4 (10d) | GMM40 (50d) | MoS (50d) |
|---|---|---|---|---|---|---|
| **SCLD** | $134.23_{\pm 8.39}$ | $0.44_{\pm 0.06}$ | $0.31_{\pm 0.04}$ | $0.40_{\pm 0.01}$ | $3787.73_{\pm 249.75}$ | $656.10_{\pm 88.97}$ |
| **SCLD-NoMCMC** | $147.38_{\pm 7.84}$ | $0.44_{\pm 0.05}$ | $0.31_{\pm 0.04}$ | $0.41_{\pm 0.01}$ | $3929.52_{\pm 753.27}$ | $1252.87_{\pm 183.95}$ |

Using the same experimental setting as the main experiments, we compare the effect of omitting SMC steps during training and evaluation in Tables 7 and 8. Unsurprisingly, removing MCMC steps has an adverse effect on performance. However, in many cases, the difference is not too big. In particular, on tasks where a smaller number of 4 subtrajectories have been used (Robot1, Robot4,

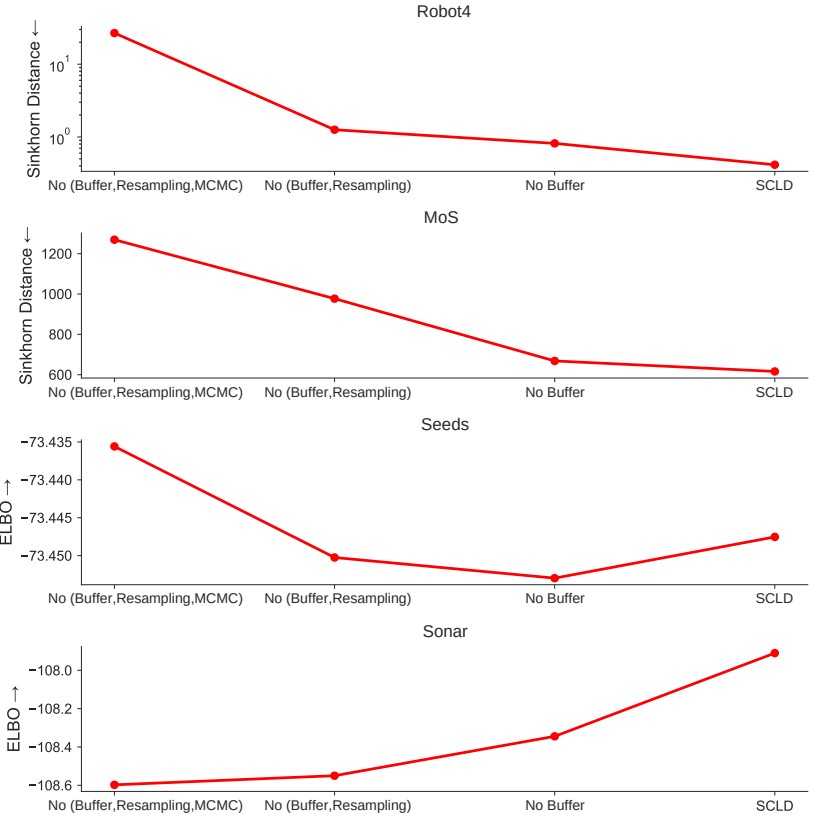

Figure 6: Ablation study of the different components of SCLD on four tasks. We sequentially add MCMC steps, resampling, and a prioritized relay buffer to LV-CMCD with subtrajectories (corresponding to the "No (Buffer,Resampling,MCMC)" method) to arrive at our proposed SCLD method. We observe that on most tasks, each of these components improves performance.

GMM40, MW54), the effect was negligible, as MCMC steps did not feature prominently in the training process in the first place. On the other tasks, where 128 SMC steps have been employed, the impact on performance was larger. However, the performance was still competitive with other approaches, noting that we did not increase the number of gradient steps. In all, using SCLD without MCMC steps is shown to be a viable possibility. It is also plausible that increased noise levels could help compensate for the lack of additional randomness.

### A.6.3 TIMINGS

In Table 9, we report the timings on each task for each of the methods in the main table with regards to time taken per gradient step (except SMC, which does not require training), using the same hyperparameters as for the main experiments. We worked in the JAX framework and used jitting, discarding the first iteration (Bradbury et al., 2018). We average across 3 seeds on a single Nvidia RTX4090 GPU for 500 iterations. Dynamical memory allocation via XLA_PYTHON_CLIENT_ALLOCATOR=platform was required for CMCD-KL on GMM40 to fit within the memory limit, resulting in slower runtimes.

Table 9: Average time per gradient step for all considered methods and tasks.

| Time (s) | Brownian | Credit | LGCP | Seeds | Sonar | Funnel | GMM40 | MW54 | Robot1 | Robot4 | MoS |
|---|---|---|---|---|---|---|---|---|---|---|---|
| **CMCD-KL** | 0.21 | 0.14 | 0.39 | 0.12 | 0.13 | 0.14 | 0.58 | 0.10 | 0.24 | 0.24 | 0.20 |
| **CMCD-LV** | 0.34 | 1.41 | 0.42 | 0.13 | 0.18 | 0.10 | 0.14 | 0.09 | 0.11 | 0.12 | 0.11 |
| **SCLD** | 0.13 | 0.15 | 1.48 | 0.07 | 0.11 | 0.07 | 0.12 | 0.07 | 0.08 | 0.08 | 0.09 |
| **CRAFT** | 0.06 | 0.004 | 0.89 | 0.03 | 0.04 | 0.02 | 0.01 | 0.02 | 0.01 | 0.05 | 0.04 |
| **DDS** | 0.04 | 0.03 | 0.11 | 0.03 | 0.03 | 0.03 | 0.04 | 0.03 | 0.04 | 0.04 | 0.03 |
| **PIS** | 0.04 | 0.03 | 0.10 | 0.03 | 0.03 | 0.03 | 0.04 | 0.03 | 0.04 | 0.04 | 0.03 |

The dimension of the target, the number of SMC operations, as well as the difficulty of evaluating the target all significantly influence the computation time. It may seem strange that SCLD, with the added complexity of SMC steps, was generally faster than the CMCD variants. This can be attributed to two points. First, SCLD detaches the trajectory due to the use of the off-policy log-variance loss, unlike CMCD-KL, which results in a simplified computation graph, saving both time and memory. We refer to Richter & Berner (2024) for a full discussion on using detaching in the log-variance loss. Due to our use of the subtrajectory-based LV loss, the gradients for each subtrajectory can be computed independently and in parallel, improving speed over CMCD-LV. Please note, however, that timings are highly dependent on implementational details.

### A.6.4 ESTIMATIONS OF THE NORMALIZING CONSTANT

When the true normalizing constant $Z$ for a density is known, another benchmark often used to evaluate a sampler is to study how accurately it can estimate $Z$ or $\log Z$. It is, however, known that for multimodal tasks, methods that achieve good $\log Z$ estimates often do so at the expense of mode collapse. Conversely, methods that avoid mode collapse sometimes yield poor $\log Z$ estimates (Blessing et al., 2024). Indeed, applying (tuned) CRAFT to the GMM40 (50d) task achieves an $\log Z$ estimate of $-3.63$ (the true value is $\log 1 = 0$) and is one of the better-performing methods for the task. While this may sound impressive, it is realized that $-3.63 \approx -\log 40$ corresponds to sampling perfectly from exactly 1 of the 40 modes (as evidenced by Figure 7). Thus while CRAFT achieves relatively good estimates of the true $\log Z$, it performs poorly as a sampler. Likewise, when SCLD is optimized for Sinkhorn distances, it often has worse estimation errors but achieves significantly better sample quality.

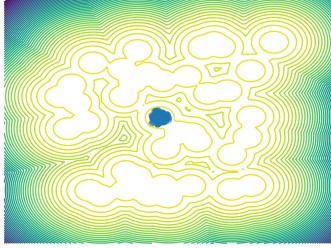

Figure 7: CRAFT only samples from one mode of GMM40 (50d).

Acknowledging this trade-off between $\log Z$ estimation and mode collapse, we present two sets of results for CRAFT, CMCD-KL, CMCD-LV, and SCLD corresponding to the $\log Z$ estimation error when methods are optimized for Sinkhorn distances (named CRAFT-SD, SCLD-SD, CMCD-KL-SD, CMCD-LV-SD) and when methods are optimized for $\log Z$ estimation (named correspondingly). In Table 10, we present errors of normalizing constant estimations on a selection of tasks where true $\log Z$ values are available, averaged over 4 seeds and using the same evaluation protocol as the main experiments. For this experiment, results for DDS and PIS are also taken from Blessing et al. (2024) when available.

Table 10: $\log Z$ estimations for different tasks.

| $\Delta \log Z$ ($\downarrow$) | Funnel (10d) | MW54 (5d) | GMM40 (2d) | GMM40 (50d) | MoS (50d) |
|---|---|---|---|---|---|
| **SMC** | $0.19_{\pm 0.09}$ | $1.45_{\pm 1.53}$ | $0.08_{\pm 0.03}$ | $761.93_{\pm 21.55}$ | $3.88_{\pm 1.76}$ |
| **PIS** | $0.92_{\pm 0.60}$ | $0.36_{\pm 0.07}$ | $0.27_{\pm 0.01}$ | $7.12_{\pm 0.63}$ | $12.25_{\pm 0.33}$ |
| **DDS** | $0.19_{\pm 0.08}$ | $3.34_{\pm 0.08}$ | $0.01_{\pm 0.01}$ | $1.74_{\pm 0.44}$ | $7.95_{\pm 0.30}$ |
| **CRAFT-SD** | $0.10_{\pm 0.02}$ | $0.16_{\pm 0.05}$ | $0.02_{\pm 0.02}$ | $6295.25_{\pm 144.71}$ | $0.75_{\pm 0.19}$ |
| **CRAFT-logZ** | $0.10_{\pm 0.02}$ | $0.16_{\pm 0.05}$ | $0.02_{\pm 0.01}$ | $3.63_{\pm 0.05}$ | $0.75_{\pm 0.19}$ |
| **CMCD-KL-SD** | $\mathbf{0.04_{\pm 0.01}}$ | $1.65_{\pm 0.10}$ | $0.01_{\pm 0.00}$ | $3.53_{\pm 0.12}$ | $2.72_{\pm 0.45}$ |
| **CMCD-KL-logZ** | $\mathbf{0.04_{\pm 0.01}}$ | $1.65_{\pm 0.10}$ | $0.01_{\pm 0.00}$ | $3.53_{\pm 0.12}$ | $2.19_{\pm 0.36}$ |
| **CMCD-LV-SD** | $0.24_{\pm 0.10}$ | $\mathbf{0.01_{\pm 0.01}}$ | $0.01_{\pm 0.00}$ | $1.45_{\pm 0.35}$ | $3.04_{\pm 0.41}$ |
| **CMCD-LV-logZ** | $0.18_{\pm 0.05}$ | $\mathbf{0.01_{\pm 0.01}}$ | $\mathbf{0.00_{\pm 0.00}}$ | $1.45_{\pm 0.35}$ | $3.04_{\pm 0.41}$ |
| **SCLD-SD** | $0.09_{\pm 0.01}$ | $0.14_{\pm 0.03}$ | $0.02_{\pm 0.01}$ | $7.10_{\pm 4.05}$ | $\mathbf{0.05_{\pm 0.03}}$ |
| **SCLD-logZ** | $0.09_{\pm 0.01}$ | $\mathbf{0.01_{\pm 0.00}}$ | $0.02_{\pm 0.01}$ | $\mathbf{0.77_{\pm 0.66}}$ | $\mathbf{0.05_{\pm 0.03}}$ |

We found that using SMC at evaluation time (with the same configuration as during training) consistently improved $\log Z$ estimate quality for SCLD and consequently used it for all tasks. We maintain the same subtrajectory settings as we did for the main experiments. SCLD significantly outperforms all other methods on the GMM40 (50d) and MoS tasks and is best or a close second on the other tasks. This illustrates that our method can also be adjusted to target better $\log Z$ estimates.

### A.6.5 THE LEARNED ANNEALING SCHEDULE

For CMCD-KL, CMCD-LV, and SCLD, we found that using a learned annealing schedule as in (35) is crucial to obtaining good results. We illustrate this in Figure 8 with a case study on SCLD, visualizing the linearly interpolated annealing schedule, i.e., (20) with $\beta(t) = t/T$, and the learned annealing schedule in (35) for the 2-dimensional GMM40 task.

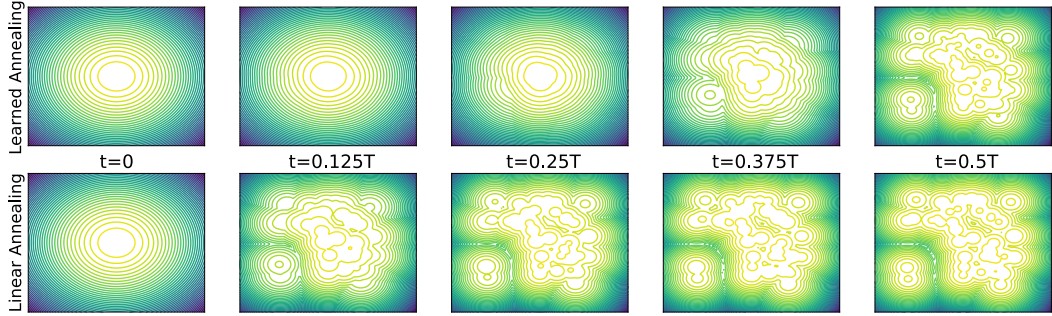

Figure 8: We compare the uniform annealing schedule with the annealing schedule learned by SCLD for $0 \leq t \leq T/2$. SCLD is able to learn a more gradual annealing schedule, which potentially allows transitions between adjacent densities to be learned more easily.

### A.6.6 MEAN FIELD PRIOR FOR SCLD

While we opt for a prior of the form $\mathcal{N}(0, \sigma^2 I)$ for SCLD in our main experiments, an alternative approach is to initialize it using a diagonal Gaussian trained using *Mean Field Variational Inference* (MFVI) (Bishop, 2006). We study this design choice experimentally here.

We use 50000 iterations of MFVI with batch size 2000 and constant learning rate $10^{-3}$, initializing with $\mathcal{N}(0, I)$. We retain the same experimental setup and hyperparameter settings as for the main experiments, except for the max diffusion coefficient, where we divide the values from the main experiments by 10. This is because MFVI is mode seeking, and so aims to cover a high probability region of the target distribution tightly, leading to a prior with smaller support. We compare the attained ELBOs in Table 11 and also report results for SCLD-MFVI at initialization (i.e., without training the control), termed "NoTrain".

Table 11: Performance of SCLD when fitting the diagonal of the prior covariance matrix using MFVI at initialization ("NoTrain") and after training ("SCLD-MFVI").

| ELBOs ($\uparrow$) | Brownian | Credit | LGCP | Seeds | Sonar |
|---|---|---|---|---|---|
| **NoTrain** | $1.07_{\pm 0.23}$ | $-513.70_{\pm 0.70}$ | $\mathbf{500.42_{\pm 0.37}}$ | $-73.48_{\pm 0.05}$ | $-114.89_{\pm 1.35}$ |
| **SCLD** | $1.00_{\pm 0.18}$ | $\mathbf{-504.46_{\pm 0.09}}$ | $486.77_{\pm 0.70}$ | $\mathbf{-73.45_{\pm 0.01}}$ | $\mathbf{-108.17_{\pm 0.25}}$ |
| **SCLD-MFVI** | $\mathbf{1.14_{\pm 0.05}}$ | $-504.59_{\pm 0.15}$ | $\mathbf{500.56_{\pm 0.12}}$ | $\mathbf{-73.44_{\pm 0.01}}$ | $-108.93_{\pm 0.34}$ |

Impressively, SCLD often achieves near-state-of-the-art results even without training when initialized with MFVI, such as on the LGCP task. We can attribute this to SCLD being initialized as an SMC sampler with Unadjusted Langevin Annealing (ULA) transition kernels as well as MCMC steps, which, in conjunction with the mode-seeking behavior of MFVI, leads to high ELBO values. SCLD-MFVI attains competitive performances on all tasks. Given that we performed no re-tuning on SCLD-MFVI, it is probable that with more careful setting and hyperparameter choices, even higher ELBOs could be attained.

However, using MFVI-fitted priors in practice often carries serious drawbacks. In line with the experiments of Blessing et al. (2024), we found that using MFVI priors leads to mode collapse

(due to the mode-seeking nature of MFVI training restricting the sampling to a subset of the target modes), and thus potentially poor sample quality. We illustrate this in Figure 9 on the GMM40 (50d) target, where we use the same hyperparameters as in the main experiment except for the prior.

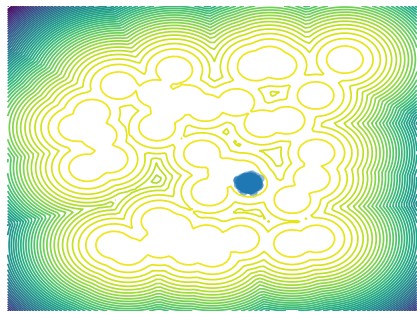

Figure 9: The samples drawn by SCLD when an MFVI-fitted prior is used. MFVI obtains a prior density that covers exactly one mode of the GMM40 distribution. As such, SCLD is unable to discover the other modes and experiences complete mode collapse. This is in contrast to Figure 2 where SCLD samples are visually indistinguishable from the target density.

### A.6.7 COMPARISON TO PDDS

In this section, we empirically compare the PDDS and SCLD methods. We employ the exact experimental methodology of Phillips et al. (2024). In particular, we train for 20000 gradient steps, refreshing the model every 500 steps. We employed 50000 gradient steps to train the mean field prior. We note that this corresponds to a significantly higher iteration budget than was allocated to SCLD. In line with the findings of Phillips et al. (2024), we found that sweeping over the prior scale as opposed to using a variational approximation significantly degraded performance on all tasks (and indeed on several tasks, such as Robot and GMM40 could not train at all). One reason for the degraded performance might be that PPDS is unable to further optimize the prior during training (as is done in SCLD). We thus opt to use variational approximations (by mean field Gaussians) to initialize the prior for all tasks. Benchmarking was done exactly as in the main experiments, and we analyzed the performance of PDDS with and without MCMC steps.

For all tasks present in the benchmark of Phillips et al. (2024) (including the Gaussian mixture tasks), we used the pre-tuned MCMC step sizes. For the other tasks, we chose a linearly interpolated step size schedule from $t = 0$ to $t = T$ where step sizes at times $0$ and $T$ are taken from the grid $[0.1, 0.3, 1, 3, 10]$ since the method for tuning MCMC step sizes was not specified. We select the best parameters directly based on the target metric and present the results in Tables 12 and 13.

Table 12: Comparison of SCLD against PDDS (Phillips et al., 2024) in terms of ELBOs.

| ELBOs ($\uparrow$) | Brownian | Credit | LGCP | Seeds | Sonar |
|---|---|---|---|---|---|
| **PDDS** | $\mathbf{1.12_{\pm0.23}}$ | $\mathbf{-502.80_{\pm0.72}}$ | $499.35_{\pm0.65}$ | $-73.48_{\pm0.21}$ | $-108.61_{\pm0.06}$ |
| **PDDS-MCMC** | $1.04_{\pm0.04}$ | $\mathbf{-502.90_{\pm0.28}}$ | $499.83_{\pm0.08}$ | $-73.47_{\pm0.19}$ | $-108.67_{\pm0.04}$ |
| **SCLD (ours)** | $1.00_{\pm0.18}$ | $-504.46_{\pm0.09}$ | $486.77_{\pm0.70}$ | $\mathbf{-73.45_{\pm0.01}}$ | $\mathbf{-108.17_{\pm0.25}}$ |
| **SCLD-MFVI (ours)** | $\mathbf{1.14_{\pm0.05}}$ | $-504.59_{\pm0.15}$ | $\mathbf{500.56_{\pm0.12}}$ | $\mathbf{-73.44_{\pm0.01}}$ | $-108.93_{\pm0.34}$ |

Table 13: Comparison of SCLD against PDDS (Phillips et al., 2024) in terms of Sinkhorn distances.

| Sinkhorn ($\downarrow$) | Funnel | GMM40 | MW54 | Robot1 | Robot4 | MoS |
|---|---|---|---|---|---|---|
| **PDDS** | $145.81_{\pm13.28}$ | $42157.92_{\pm346.21}$ | $1.28_{\pm0.18}$ | $3.36_{\pm0.08}$ | $3.09_{\pm0.16}$ | $3119.83_{\pm98.64}$ |
| **PDDS-MCMC** | $151.02_{\pm28.00}$ | $42157.92_{\pm346.21}$ | $1.07_{\pm0.25}$ | $3.35_{\pm0.08}$ | $3.08_{\pm0.14}$ | $3108.75_{\pm98.61}$ |
| **SCLD (ours)** | $\mathbf{134.23_{\pm8.39}}$ | $\mathbf{3787.73_{\pm249.75}}$ | $\mathbf{0.44_{\pm0.06}}$ | $\mathbf{0.31_{\pm0.04}}$ | $\mathbf{0.40_{\pm0.01}}$ | $\mathbf{656.10_{\pm88.97}}$ |

PDDS attains comparable ELBOs to SCLD on the Bayesian statistics tasks. This is due to both methods being initialized as SMC samplers with a prior obtained by the same variational approximation

(for SCLD-MFVI). We also observed, in line with the findings of Phillips et al. (2024) and similar to Appendix A.6.6, that often relatively little training is required to achieve optimal performance, so the gap in performance between the initial, untrained SMC scheme and the trained sampler is small.

However, PDDS consistently presents significantly worse Sinkhorn distances (on all tasks where this is available) than SCLD. This is due to the reliance of PDDS on using an MFVI prior, which, as discussed in Appendix A.6.6, is prone to mode collapse. On the other hand, SCLD is able to operate stably without relying on using the MFVI prior, avoiding mode collapse.

### A.6.8 COMPARISON WITH ADVANCED SMC SCHEMES

In the section, we compare SCLD against two advanced SMC schemes implemented in the framework by Cabezas et al. (2024). We consider *adaptive tempered SMC*, which utilizes the *constant-ESS* method for choosing the annealing schedule as seen in Buchholz et al. (2021). We term this method SMC-ESS. In line with SCLD, we utilize a single HMC step for the SMC kernel with 10 leapfrog integration steps, and apply the same tuning procedure for HMC step size as we did for our own SMC method. We additionally sweep over the ESS threshold $\alpha \in \{0.3, 0.5, 0.75, 0.9, 0.95, 0.99\}$. Due to the large search grid, we run 10 seeds per task to mitigate outliers. Unlike SCLD, which uses multinomial resampling (for a fair comparison to our other baselines), we use systematic resampling (see, e.g., Chopin et al. (2020, Chapter 9)) for SMC-ESS, which we found led to best performance. We consider another method from Buchholz et al. (2021), utilizing the *full-covariance tuning* approach for *Independent Rosenbluth Metropolis-Hastings* (IRMH) proposals (on top of using adaptive tempered SMC). We use 100 MCMC steps per step and term this method SMC-FC.

We report results in Tables 2 and 3, using the same evaluation protocol (in particular, using 2000 particles). For reference, we also compare all SMC methods with SCLD in Tables 14 and 15.

Table 14: Comparison of SCLD against advanced SMC methods (Buchholz et al., 2021) in terms of ELBOs.

| ELBOs ($\uparrow$) | Brownian | Credit | LGCP | Seeds | Sonar |
|---|---|---|---|---|---|
| **SMC** | $-2.21_{\pm 0.53}$ | $-589.82_{\pm 5.72}$ | $385.75_{\pm 7.65}$ | $-74.63_{\pm 0.14}$ | $-111.50_{\pm 0.96}$ |
| **SMC-ESS** | $0.49_{\pm 0.19}$ | $-505.57_{\pm 0.18}$ | $\mathbf{497.85_{\pm 0.11}}$ | $-74.07_{\pm 0.60}$ | $-109.10_{\pm 0.17}$ |
| **SMC-FC** | $-1.91_{\pm 0.04}$ | $-505.30_{\pm 0.02}$ | $-878.10_{\pm 2.20}$ | $-74.07_{\pm 0.02}$ | $-108.93_{\pm 0.02}$ |
| **SCLD (ours)** | $\mathbf{1.00_{\pm 0.18}}$ | $\mathbf{-504.46_{\pm 0.09}}$ | $486.77_{\pm 0.70}$ | $\mathbf{-73.45_{\pm 0.01}}$ | $\mathbf{-108.17_{\pm 0.25}}$ |

Table 15: Comparison of SCLD against advanced SMC methods (Buchholz et al., 2021) in terms of Sinkhorn distances.

| Sinkhorn ($\downarrow$) | Funnel | GMM40 | MW54 | Robot1 | Robot4 | MoS |
|---|---|---|---|---|---|---|
| **SMC** | $149.35_{\pm 4.73}$ | $46370.34_{\pm 137.79}$ | $20.71_{\pm 5.33}$ | $24.02_{\pm 1.06}$ | $24.08_{\pm 0.26}$ | $3297.28_{\pm 2184.54}$ |
| **SMC-ESS** | $\mathbf{117.48_{\pm 9.70}}$ | $24240.68_{\pm 50.52}$ | $1.11_{\pm 0.15}$ | $1.82_{\pm 0.50}$ | $2.11_{\pm 0.31}$ | $1477.04_{\pm 133.80}$ |
| **SMC-FC** | $211.43_{\pm 30.08}$ | $39018.27_{\pm 159.32}$ | $2.03_{\pm 0.17}$ | $0.37_{\pm 0.08}$ | $1.23_{\pm 0.02}$ | $3200.10_{\pm 95.35}$ |
| **SCLD (ours)** | $134.23_{\pm 8.39}$ | $\mathbf{3787.73_{\pm 249.75}}$ | $\mathbf{0.44_{\pm 0.06}}$ | $\mathbf{0.31_{\pm 0.04}}$ | $\mathbf{0.40_{\pm 0.01}}$ | $\mathbf{656.10_{\pm 88.97}}$ |

The full-covariance tuning and the ESS-based scheme for selecting the annealing schedule significantly outperform our baseline implementation of SMC at the expense of longer and variable (possibly unbounded) sampling times. Nevertheless, all considered SMC methods are superseded by SCLD in performance on all but two tasks. While SCLD uses a relatively simple version of SMC for fair comparisons to our baselines, our framework enables the usage of more advanced techniques, such as those used for SMC-ESS and SMC-FC. Thus, we expect that the performance of SCLD can be even further improved.

### A.6.9 CONVERGENCE OF DIFFERENT METHODS BY ITERATION COUNT

In Figure 10, we visualize the same data as in Section 3.1 but plotting by the number of elapsed gradient steps. In this perspective, the same conclusions hold that SCLD exhibits superior convergence properties, attaining the best ELBOs on each task for all numbers of gradient steps. Note that CRAFT was not competitive on the Credit task in this perspective.

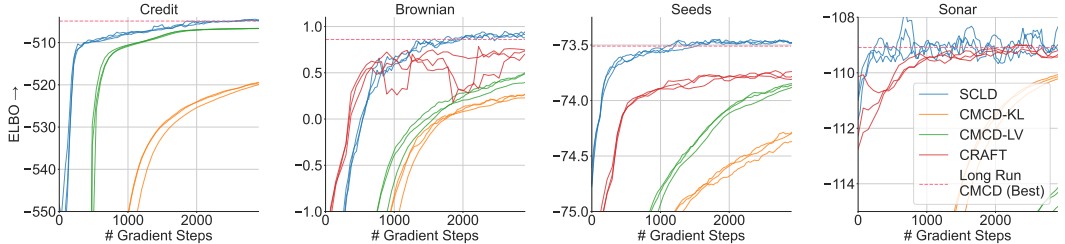

Figure 10: The same experiments as in Figure 3 plotted instead by iterations.

### A.6.10 KL-BASED TRAINING OF SCLD

We compare KL and LV-based training of the SCLD algorithm, using the family of Funnel distributions with $d \in \{10, 20, 30, 40, 50\}$ as a case study. We train SCLD using KL and LV losses with $4$ and $128$ subtrajectories as described in Section 2.3 for 3000 gradient steps using the same hyperparameters and settings (including learning the annealing schedule and prior) as in the $d = 10$ case for the main experiments. In Figure 11, we visualize the ELBOs attained (using the same settings during evaluation as for training) alongside CMCD-KL and CMCD-LV.

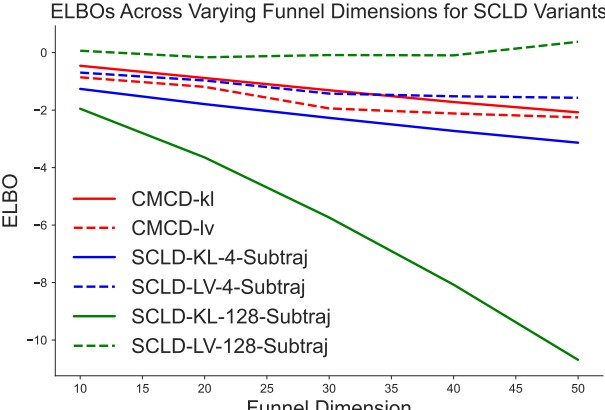

Figure 11: ELBOs across Varying Funnel Dimensions for different SCLD-variants

All methods except SCLD with LV loss and $128$ subtrajectories experience some form of performance degradation as dimensions scale. For the LV loss, adding subtrajectories reduces the amount of performance degradation. This may be due to SMC steps countering increased dimensionality by focusing computation on high-density regions. For the KL loss, however, increasing the number of subtrajectories resulted in worse performance, especially as the dimension increased. Indeed, SCLD-KL with $128$ subtrajectories scales the most poorly of the methods tried as $d$ increases. As discussed in Section 2.3, this may be due to the use of importance sampling to estimate the loss function. Indeed, a set of importance weights is required for each subtrajectory to estimate the loss, and thus, using more subtrajectories demands a greater reliance on importance sampling. As the variance of importance sampling can increase significantly with dimension, this may account for the decreased performance of KL-based subtrajectory losses. In summary, this supports the hypothesis that losses avoiding importance sampling, such as the log-variance loss, are more suited to the training of SCLD on higher-dimensional tasks.

