# OpenReview forum: "Sequential Controlled Langevin Diffusions"
_ICLR.cc/2025/Conference — ICLR 2025 Poster_

### Official Review · Reviewer_ZuHY · 2024-10-23

**Soundness:** 3
**Presentation:** 3
**Contribution:** 2
**Rating:** 6
**Confidence:** 4

**Summary:**

This paper proposes a new algorithm for sampling from unnormalized distributions. The core idea is to combine the capacity of SMC to focus on relevant regions of the space with the ability of diffusion based samplers to adapt to the geometry of the target distribution by learning a control term $u$ of the reverse SDE.


The control term $u$ then parametrizes a divergence between the path space measure of the forward process and the measure of the backward process.
By minimizing this divergence the algorithm ensures that the two processes become aligned.


To learn the optimal control $u$ the optimization objective is split on N intervals $[t_n, t_{n+1}]$ over which the divergence can be estimated by running the SMC procedure.
The learned control $u$ is then used to guide the proposals in the SMC sampler.

The method is benchmarked against recent approaches on standard benchmarks.

**Strengths:**

The paper makes a good job at identifying the strengths and weaknesses of SMC samplers and of diffusion based samplers and how they can be combined together. The main concepts needed to understand the method are well introduced and the paper does a good job at explaining them.

**Weaknesses:**

A significant portion of the paper focuses on presenting the foundational concepts and prior research upon which SCLD is constructed. While this provides necessary context, very little space is used for a detailed discussion of the novel aspects of the sampler itself. As a result, the reader struggles to properly get the contributions.

A comparisons with more traditional SMC samplers to justify the need for the control term and the training procedure would be a nice thing to have. For example with the HMC kernel adaptive procedure of [1] or the adaptive tempered version of [3] (both already implemented in [2]).

In SMC, the choice of the time discretization is usually motivated by the need to have a constant effective sample size (ESS) per step.
This is quite critical for good sampler performance and principled ways like [3] are available to achieve this. When moving to the continuous time/path space measure setting of SCLD it seems to me that we loose these principled ways to choose the time discretization. It is worth comparing if the flexibility gained in the proposal outweighs the loss of this machinery.
Or if there is a principled way to choose the time discretization for the sampling process of SCLD.


[1] - Buchholz, A., Chopin, N. and Jacob, P.E., 2021. Adaptive tuning of hamiltonian monte carlo within sequential monte carlo. Bayesian Analysis
[2] - https://github.com/blackjax-devs/blackjax
[3] - Jasra A, Stephens DA, Doucet A, Tsagaris T,  2011. Inference for Lévy‐driven stochastic volatility models via adaptive sequential Monte Carlo. Scandinavian Journal of Statistics.

**Questions:**

- Are there clear examples where the more flexible proposal is worth the added need for training over running a slightly more sophisticated SMC sampler (eg [1], [3])?

- Are there more complex and less toy examples where SCLD would be applicable ?

-  How sensitive is the procedure wrt the chosen time discretization $[t_n, t_{n+1}]$?

- When sampling, is it possible for SCLD to maintain a principled time discretization guidance similar to traditional SMC use of constant ESS? Does the flexibility in proposal distribution come at the cost of losing these methods?  If so, how does this trade-off impact the overall performance and reliability of the sampler?

---

> ### Author Response · Authors · 2024-11-20
> **Response to reviewer ZuHY (1/3)**
>
> Dear Reviewer ZuHY,
>
> Thank you for your considered review. We are pleased that you appreciate our pedagogical style of writing, and are very grateful for your added insights, especially on the constant ESS idea. We respond to your questions and raised issues below.
>
> ## 1. The reader struggles to properly get the contributions
>
> We provide several novel ideas in our paper. We introduce a new SMC framework for combining SMC with diffusion samplers by adopting a continuous path space perspective to SMC and formulating a novel family of training objectives, culminating in the highly effective Sequential Controlled Langevin Diffusion (SCLD) method. Please refer to our General Response for a more detailed discussion of our contributions.
>
> ## 2. A comparison with more traditional SMC samplers to justify the need for the control term and the training procedure would be a nice thing to have
>
> We will jointly answer this in the next point.
>
> ## 3. Are there clear examples where the more flexible proposal is worth the added need for training over running a slightly more sophisticated SMC sampler (eg [1], [3])?
>
> We compare SCLD against the Adaptive tempered SMC implemented in [2], which utilizes the constant ESS method for choosing discretization steps as seen in [1], [3] (we identify it as the closest relative among the Blackjax library of the methods presented). As for SCLD, we utilize a single HMC step for the SMC kernel with 10 leapfrog integration steps, and apply the same tuning procedure for step size as we did for our own SMC method. We additionally sweep over the ESS threshold $\alpha \in \{0.3, 0.5, 0.75, 0.9, 0.95, 0.99\}$ and use systematic resampling. Otherwise, we apply the same benchmarking procedure as for our other experiments.
>
> We have added a section (A6.8, in orange) in the appendices of our paper concerning this comparison. While the new SMC implementation (which we refer to as SMC-ESS) significantly outperforms the baseline, it is still worse than SCLD on 9 out of 11 benchmark tasks, while also sacrificing a key benefit of SCLD which is that of fixed sampling times once trained. For instance, SMC-ESS was only able to attain better performance on LGCP in part due to taking 789 SMC steps on average, compared to SCLD's 128. We display the results below for reference as well.
>
> ### Table: Comparison of SCLD against SMC-ESS in terms of ELBOs
> Comparison of SCLD against SMC-ESS in terms of ELBOs.
>
> | ELBOs ($\uparrow$)   | Brownian           | Credit             | LGCP               | Seeds             | Sonar             |
> |-----------------------|--------------------|--------------------|--------------------|-------------------|-------------------|
> | **SMC**              | $-2.21 \pm 0.53$ | $-589.82 \pm 5.72$ | $385.75 \pm 7.65$ | $-74.63 \pm 0.14$ | $-111.50 \pm 0.96$ |
> | **SMC-ESS**          | $0.49 \pm 0.19$  | $-505.57 \pm 0.18$ | $\pmb{497.85 \pm 0.11}$ | $-74.07 \pm 0.60$ | $-109.10 \pm 0.17$ |
> | **SCLD (ours)**      | $\pmb{1.00 \pm 0.18}$ | $\pmb{-504.46 \pm 0.09}$ | $486.77 \pm 0.70$ | $\pmb{-73.45 \pm 0.01}$ | $\pmb{-108.17 \pm 0.25}$ |
>
> ---
>
> ### Table: Comparison of SCLD against SMC-ESS in terms of Sinkhorn distances
> Comparison of SCLD against SMC-ESS in terms of Sinkhorn distances.
>
> | Sinkhorn ($\downarrow$) | Funnel           | GMM40             | MW54              | Robot1           | Robot4           | MoS              |
> |--------------------------|------------------|--------------------|-------------------|------------------|------------------|------------------|
> | **SMC**                 | $149.35 \pm 4.73$ | $46370.34 \pm 137.79$ | $20.71 \pm 5.33$ | $24.02 \pm 1.06$ | $24.08 \pm 0.26$ | $3297.28 \pm 2184.54$ |
> | **SMC-ESS**             | $\pmb{117.48 \pm 9.70}$ | $24240.68 \pm 50.52$ | $1.11 \pm 0.15$ | $1.82 \pm 0.50$ | $2.11 \pm 0.31$ | $1477.04 \pm 133.80$ |
> | **SCLD (ours)**         | $134.23 \pm 8.39$ | $\pmb{3787.73 \pm 249.75}$ | $\pmb{0.44 \pm 0.06}$ | $\pmb{0.31 \pm 0.04}$ | $\pmb{0.40 \pm 0.01}$ | $\pmb{656.10 \pm 88.97}$ |
>
> Finally, we note that such improved SMC methods can also be used for SCLD to further improve its performance. We will discuss this in a subsequent section.
>
> (response continued in next post)

---

> ### Author Response · Authors · 2024-11-20
> **Response to reviewer ZuHY (2/3)**
>
> ## 4. Are there more complex and less toy examples where SCLD would be applicable?
>
> We first emphasize that our tasks are considered challenging benchmarks, covering up to $1600$-dimensional densities from Bayesian inference, $50$-dimensional multimodal densities with $40$ modes, as well as problems from robotic navigation where most existing methods fail. We would say that these are arguably among the most complex tasks used to benchmark other samplers, such as those presented in our related work section.
>
> Nevertheless, we want to illustrate the application of SCLD to the simulation of a statistical lattice field theory. We study the lattice $\phi^4$ theory in $D = 2$ spacetime dimensions (however, note that the dimension of our density will be $d=128$, see below). The random variables in this circumstance are field configurations $\phi \in \mathbb{R}^{W\times L}$, and their density is given by
>
> $$p_{\mathrm{target}}(\phi)=\frac{e^{-U(\phi)}}{Z},$$
>
> where $$U(\phi)=-2\kappa\sum_x\sum_{\mu}\phi_x\phi_{x+\mu}\ +\ (1-2\lambda)\sum_x \phi_x^2 \ +\ \lambda \sum_x \phi_x^4$$
>
> and where summation over $x$ denotes summation over lattice sites and summation over $\mu$ corresponds to summing over neighbours. For our experiments we pick $W=16$ and $L=8$, leading to a target in $d=16\cdot 8 = 128$ dimensions. We also pick $\lambda=0.022$ and $\kappa=0.3$ (corresponding to a so-called *critical point*) as in [c].
>
> A physical question of interest is to estimate the free energy $F=-\frac{1}{d}\log Z$ of the system. We used long-run HMC as in [c] to obtain groundtruth free energy values. We apply both the SMC-ESS method considered previously and SCLD. We tune adaptive tempered SMC (SMC-ESS) in the same manner as before, maximizing ELBOs. For SCLD we use the same design choices as we did for the main experiments except for the network, where we use an equivariant version of PISGradNet that satisfies $u(\pmb{x})=-u(-\pmb{x})$ to account for the symmetry in the potential. For both SCLD and SMC-ESS we estimate the free energy with 2000 particles.
>
> We run SCLD with 4 training seeds for 2000 training iterations and evaluate both SCLD and SMC-ESS using 50 evaluation seeds. We compute the average error $\Delta F$ of our free energy estimates across seeds and obtain the following results:
>
>
> | Method  | $\Delta F$ ($\downarrow$)           |
> | ------- | --------------------- |
> | SCLD    | $\pmb{0.00269\pm 0.000286}$ |
> | SMC-ESS | $0.00299\pm 0.000636$ |
>
> SCLD is able to obtain on average better estimates of free energy than SMC-ESS. We will add a comparison to all baselines in our final version. We anticipate that the application of SCLD to further problems in the natural sciences is a promising area of future work.
>
>
> ## 5. When moving to the continuous time/path space measure setting of SCLD it seems to me that we loose these principled ways to choose the time discretization
>
> We assume here that "time discretization" for traditional discrete-time SMC algorithms refers to the chosen annealing schedule of the densities (i.e the $\lambda_t$ in [1], which we denote $\beta_t$ in our paper). We will refer to this subsequently by the annealing schedule. In our algorithm, we would like to highlight that our choice of $\beta_t$ (which determines factors like ESS) is entirely decoupled from how we discretize time.
>
> In SCLD, we are able to, and choose to, learn $\beta_t$ (Section 2.4 - Annealing path) via direct gradient optimization of the divergence objective. Gradient optimization is a principled and desirable (see [a]) way of selecting algorithm parameters, so while we don't choose the $\beta_t$ with ESS in mind, our method of choosing the annealing schedule is also principled. In fact, our method of choosing $\beta_t$ exhibits several benefits, the most notable being that we can draw samples using a fixed, finite number of sampling steps, which is not the case with ESS-based discretization schemes. We have added a section in the appendices (A6.5, in orange) which visually compares the learned annealing schedule with a fixed, linear annealing schedule, and shows that the learned annealing schedule is able to produce a more gradual transition from the prior to target density.
>
> We will also talk more specifically about choosing time discretization in the next point.
>
> (response continued in next post)

---

> ### Author Response · Authors · 2024-11-20
> **Response to reviewer ZuHY (3/3)**
>
> ## 6. How sensitive is the procedure wrt the chosen time discretization $[t_n, t_{n+1}]$?
>
> We also would like to note that for our main experiments we did not tune the time discretization, and the baseline choice of a cosine noise schedule with a uniform discretization is a robust choice that allowed SCLD to achieve the best performances on most of the benchmarked tasks.
>
> We would like to additionally emphasize that the choice of time discretization is somewhat redundant as a hyperparameter. A key observation is that CMCD and SCLD depends on time increments $\Delta t$ (note we use $\Delta t=\frac{t_n-t_{n-1}}{L}$ in our experiments) through $\sigma(t)\sqrt{\Delta t}$ after reparametrizing the control (see equations 32-33 in the paper). As such, any tuning of the time discretization (for a fixed step count) can be replaced with equivalent adjustments of the noise schedule $\sigma(t)$. Indeed, we can confirm this experimentally, with training trajectories remaining unchanged when rescaling time and noise levels accordingly.
>
> As such, we do not lose performance by fixing the time discretization, and it is sufficient to just tune the noise schedule. While we expect further performance improvements by further finetuning of the diffusion schedule, we as mentioned leveraged a fixed cosine schedule which has also proven successful in many previous situations such as [d, e].
>
> ## 7. When sampling, is it possible for SCLD to maintain a principled time discretization guidance similar to traditional SMC use of constant ESS? Does the flexibility in proposal distribution come at the cost of losing these methods? If so, how does this trade-off impact the overall performance and reliability of the sampler?
>
> - We have addressed the first question previously in that we can choose the annealing schedule in a principled way by simply learning it end-to-end via gradient optimization
>
> - While we do not think that switching from ESS-based ways of choosing annealing schedules to learning the annealing schedule constitutes a disadvantage, we note that as choosing the annealing schedule (such as by considering ESS) is independent of learning the control, it is also possible to choose the annealing schedule via ESS: First, we set $T=1$ and $\beta_t=t$. Then we can dynamically choose the time discretization $t_n$ based on ESS and learn a drift function that depends on the current timestep $t$ and position $x$. Note that in this case, the total number of discretization steps is variable. We can then apply log-variance-based training in the same fashion as Algorithm 2 from the paper. This constitutes an area for future investigation.
>
> ---
>
> **References**
>
> [a] MCMC Variational Inference via Uncorrected Hamiltonian Annealing: https://arxiv.org/pdf/2107.04150
>
> [b] Langevin Diffusion Variational Inference: https://arxiv.org/abs/2208.07743
>
> [c] Estimation of Thermodynamic Observables in Lattice Field Theories with Deep
> Generative Models: https://arxiv.org/pdf/2007.07115
>
> [d] Improved Denoising Diffusion Probabilistic Models https://arxiv.org/pdf/2102.09672
>
> [e] Denoising Diffusion Samplers https://arxiv.org/pdf/2302.13834

---

> > ### Comment · Reviewer_ZuHY · 2024-11-24
> >
> > I want to thank the authors for the extensive work done in order to reply to my concerns.
> >
> > I appreciated the comparison with SMC-ESS. I note that the authors included it in the Appendix. I think this paper, and the ML sampling community, could benefit from moving it to the main body of the paper. Maybe by replacing SMC by SMC-ESS if the concern is space as, in my opinion, a comparison with vanilla SMC brings very little information.
> > Also, how many particles were used for SMC-ESS ? I haven't found this information in the updated version.
> > Note also that for the SMC-ESS benchmark provided, the HMC kernel is fixed across all iterations of SMC which notoriously lead to sub-perfomance. As it is was already implemented in the framework used by the authors, I would have like to see a comparison with [1] (https://blackjax-devs.github.io/sampling-book/algorithms/TemperedSMCWithOptimizedInnerKernel.html)
> >
> > Overall I have decided to keep my score. As the method SCLD stands, I am not convinced that the more flexible proposal is worth the added complexity and need for training over running a slightly more sophisticated SMC sampler.
> >
> > I noted the author comment "While SCLD uses a relatively simple version of SMC for fair comparisons to our baselines, our framework enables the usage of more advanced techniques, such as those used for SMC-ESS. Thus, we expect that the
> > performance of SCLD can be even further improved." I think this has great potential and would encourage the authors to further explore this direction.

---

> ### Author Response · Authors · 2024-11-25
>
> Dear Reviewer ZuHY,
>
> Thank you for your response and your additional feedback. We are pleased that you recognize that SCLD and its possible derivative approaches "have great potential."
>
> **SMC experiments:** We are very sorry to hear that our additional experiments and revised paper could not convince you. We agree with your suggestion on the presentation and added the SMC-ESS results (and our new SMC-FC results outlined below) to our main tables.
>
> We would also like to emphasize that the HMC kernel for SMC-ESS is not fixed across all iterations of SMC but is instead determined by the chosen annealing schedule. We define step sizes $\varepsilon_t = f(\beta_t)$, where $\beta_t$ is the annealing schedule and $f$ is a piecewise linear function over $[0,1]$ which we select using a large grid search. As such, in terms of experimental fairness and the original review, we consider SMC-ESS to be our best interpretation of the baseline that was first requested.
>
> Nevertheless, we are happy to present an additional baseline (SMC-FC) that is taken as is from [1] (as you suggested). SMC-FC utilizes the full-covariance tuning approach for Independent Rosenbluth Metropolis-Hastings (IRMH) proposals (on top of using adaptive tempered SMC). We employ $100$ MCMC steps per step and use the same evaluation protocol as for the other methods. In particular, we use $2000$ particles for all methods (which we clarified in the appendix). We tabulate the results below.
>
> **Table: Comparison of SCLD against various SMC methods in terms of ELBOs**
>
> | ELBOs ($\uparrow$)   | Brownian           | Credit             | LGCP               | Seeds             | Sonar             |
> |-----------------------|--------------------|--------------------|--------------------|-------------------|-------------------|
> | **SMC**              | $-2.21 \pm 0.53$ | $-589.82 \pm 5.72$ | $385.75 \pm 7.65$ | $-74.63 \pm 0.14$ | $-111.50 \pm 0.96$ |
> | **SMC-ESS**          | $0.49 \pm 0.19$  | $-505.57 \pm 0.18$ | $\pmb{497.85 \pm 0.11}$ | $-74.07 \pm 0.60$ | $-109.10 \pm 0.17$ |
> | **SMC-FC**          | $-1.91 \pm 0.04$  | $-505.30 \pm 0.02$ | $-878.10 \pm 2.20$ | $-74.07 \pm 0.02$ | $-108.93 \pm 0.02$ |
> | **SCLD (ours)**      | $\pmb{1.00 \pm 0.18}$ | $\pmb{-504.46 \pm 0.09}$ | $486.77 \pm 0.70$ | $\pmb{-73.45 \pm 0.01}$ | $\pmb{-108.17 \pm 0.25}$ |
>
> ---
>
> **Table: Comparison of SCLD against various SMC methods in terms of Sinkhorn distances**
>
> | Sinkhorn ($\downarrow$) | Funnel           | GMM40             | MW54              | Robot1           | Robot4           | MoS              |
> |--------------------------|------------------|--------------------|-------------------|------------------|------------------|------------------|
> | **SMC**                 | $149.35 \pm 4.73$ | $46370.34 \pm 137.79$ | $20.71 \pm 5.33$ | $24.02 \pm 1.06$ | $24.08 \pm 0.26$ | $3297.28 \pm 2184.54$ |
> | **SMC-ESS**             | $\pmb{117.48 \pm 9.70}$ | $24240.68 \pm 50.52$ | $1.11 \pm 0.15$ | $1.82 \pm 0.50$ | $2.11 \pm 0.31$ | $1477.04 \pm 133.80$ |
> | **SMC-FC**             | $211.43 \pm 30.08$ | $39018.27 \pm 159.32$ | $2.03 \pm 0.17$ | $0.37 \pm 0.08$ | $1.23 \pm 0.02$ | $3200.10 \pm 95.35$ |
> | **SCLD (ours)**         | $134.23 \pm 8.39$ | $\pmb{3787.73 \pm 249.75}$ | $\pmb{0.44 \pm 0.06}$ | $\pmb{0.31 \pm 0.04}$ | $\pmb{0.40 \pm 0.01}$ | $\pmb{656.10 \pm 88.97}$ |
>
> ---
>
> We definitely agree that sophisticated SMC samplers have the potential to offer strong performance. However, while we see improvements of SMC-FC over SMC-ESS for several tasks (Robot1, Robot4, Credit, Sonar), SCLD still exhibits an overall advantage on 9 out of 11 tasks.
>
> **Concluding remarks:** We would like to reiterate that our main contribution is a general continuous-time framework for end-to-end learning of diffusion-based proposals. We understand the great advantages of SMC methods and *embrace* them in our work, offering a simple way to merge them with diffusion-based samplers. While we avoid more sophisticated design choices to have a fair comparison to previous methods, such as CRAFT and PDDS, we also note that there is much potential in applying more complex SMC ideas to our framework. We significantly outperform previous deep learning-based sampling methods (in terms of cost and accuracy), and as such believe our work has a place in the literature of this growing field.
>
> ---
>
> We have already revised our paper to reflect the new experiments above. We thank you again for your suggestions and hope that this response clarifies your remaining concerns.

---

> > ### Comment · Reviewer_ZuHY · 2024-11-25
> >
> > I thank the authors again for taking into account my remarks and for the additive work done.
> >
> > I am sorry I missed the grid search done for $f$ in the SMC-ESS benchmark. Thank you for benchmarking SMC-FC.
> > The tutorial I sent was for IRMH, but my point was that it can be used for tuning the HMC kernel instead (which is what is happening in the reference [1] I mentioned). It is interesting to see how a simple IRMH kernel can perform compared to deep-learning based methods when correctly tuned. I would expect a tuned HMC to perform better.
> >
> > I understand your contributions as a general continuous-time framework for end-to-end learning of diffusion-based proposals and I **note that you outperform previous deep learning-based sampling methods**.
> >
> > My point is that now that the deep learning-based sampling methods have been explored for some years, new work should think about how to bridge these to more classical SMC machinery in order to fully exploit SMC potential.
> > Or to present clear examples where the added complexity and need for training is necessary and worth it.

---

> > > ### Author Response · Authors · 2024-11-25
> > >
> > > We are pleased that we have answered your recent questions and that you recognize our core contributions, such as our general framework and SCLD's practical performance.
> > >
> > > We would like to work with you to maximize the scientific value of our work. To this end, we have now run two involved SMC baselines, SMC-ESS and SMC-FC, at your request based on our best interpretation of the communicated specifications (as you pointed out, the tutorial you sent was for IRMH, and that was indeed our best interpretation of your suggestion in the previous message). We showed that in both cases, SCLD is better, and we are grateful for your acknowledgment of our efforts on this. As the review period is coming to an end, it is increasingly unlikely that we will have the time to run any further baselines. We would be more than happy to include further baselines for a camera-ready version should the paper be accepted. We would also be able to study combining SCLD with advanced SMC techniques in this case.
> > >
> > > While we agree that *deep-learning-based* approaches for sampling have been explored for several years, we would like to emphasize that the field of *diffusion-based* sampling is still relatively young. In particular, the formulations that our work builds upon were only developed this year (see Richter and Berner, 2024, and Vargas et al., 2024).
> > >
> > > Lastly, we want to point out that our current work already seems to advance the current state in both of your suggested directions: First, we "bridge" recent deep learning-based samplers, i.e., diffusion-based samplers, to "more classical SMC machinery" through our continuous-time framework. Second, we present a wide range of more challenging benchmarks (e.g., Robot and $\phi^4$) and also analyze the accuracy vs. cost tradeoff in our convergence plots. In particular, this shows that the (already strong) initialization of SCLD, corresponding to an SMC method with ULA proposals, is continuously improved during training.
> > >
> > > We hope this helps to convince you of the value of our contributions.

---

> > > > ### Comment · Reviewer_ZuHY · 2024-11-30
> > > >
> > > > I have update my rating to reflect the extra experiments done that I think are a good added value for the ML sampling community.

---

### Official Review · Reviewer_TwR8 · 2024-10-27

**Soundness:** 3
**Presentation:** 3
**Contribution:** 2
**Rating:** 6
**Confidence:** 3

**Summary:**

This paper proposes Sequential Controlled Langevin Diffusion (SCLD), a novel sampling framework that combines the strengths of Sequential Monte Carlo (SMC) and diffusion-based sampling methods. The goal is to leverage the resampling efficiency of SMC while taking advantage of the adaptive flexibility of diffusion models. The authors further introduce a log-variance divergence loss function to address the variance issues often encountered with KL divergence estimators, aiming for more stable optimization.

**Strengths:**

- The integration of SMC and diffusion-based sampling represents a novel and promising approach to tackling challenges in high-dimensional sampling.
- The paper is generally well-structured, with clear segmentation between theory, algorithmic details, and experimental evaluation.
- The proposed log-variance divergence loss function could have meaningful applications in high-dimensional probabilistic modeling, particularly for tasks prone to mode collapse and instability.

**Weaknesses:**

While the proposed method is conceptually reasonable, the writing is quite sloppy. It is hard for me to understand the contribution and advantages of the proposed SCLD.
- The authors claim that the KL divergence estimator suffers from exponential error growth (Proposition 2.4) and the authors instead use log-variance divergence (Equation (18)). However, there is no formal proof or quantitative scaling analysis of this claim using the log-variance divergence.
- The use of "off-policy training" seems inappropriate and underexplained. This term is typically associated with reinforcement learning rather than Langevin dynamics and diffusion models.
- Section 1.1 is loosely connected to the content presented in Table 1. For example, the Particle Denoising Diffusion Sampler (PDDS) is mentioned in the appendix without proper discussion in Section 1.1. The differences between existing methods and SCLD are not sufficiently emphasized. Furthermore, the sentence "limiting them to normalizing flows, as well as MCMC steps to keep sample diversity after resampling" (lines 125–126) is unclear and requires revision.
- The log-variance divergence in Equation (18) appears to be a critical component of SCLD, but it is missing from the main Algorithm 1.

**Questions:**

- Could the authors elaborate on the finite-time convergence analysis of SCLD mentioned in Table 1? Unfortunately, I did not find a related analysis in the submission.
- The method claims to use a continuous-time setting inspired by Langevin diffusion, but the implementation appears to discretize time with fixed intervals (section 2.4). Could the authors clarify how the continuous-time formulation benefits the analysis or experimental results? How does it compare to related methods such as CMCD from [1]?

[1] Transport meets Variational Inference: Controlled Monte Carlo Diffusions. ICLR 2024

---

> ### Author Response · Authors · 2024-11-20
> **Response to reviewer TwR8**
>
> Dear Reviewer TwR8,
>
> Thank you very much for your extensive review. We appreciate that you value our novel sampling framework that integrates SMC and diffusion-based sampling and achieves state-of-the-art performance on high-dimensional examples. We are in particular happy that you highlight our application of the log-variance divergence, which indeed is an essential ingredient that has not been used for resampling-based algorithms before. Let us address your questions and comments in the sequel.
>
> ## Contribution
>
> We provide several novel ideas in our paper. We introduce a new SMC framework for combining SMC with diffusion samplers by adopting a continuous path space perspective to SMC and formulating a novel family of training objectives, culminating in the highly effective Sequential Controlled Langevin Diffusion (SCLD) method. Please refer to our General Response for a more detailed discussion of our contributions.
>
> ## Scaling analysis of log-variance divergence
>
> Thank you for raising the scaling properties of the log-variance divergence, which are indeed very relevant to our algorithm. Contrary to the KL divergence, which scales exponentially in the dimension (see Proposition 2.4), the relative error of the log-variance divergence can be shown to not have this unfavorable scaling behavior. This has been investigated in a slightly different setting already in Proposition 5.7 in [1]. For convenience, we have stated the proof in our revised version, adapted to our setting, see Appendix A.2 (highlighted in green). In addition, we compare the performance of KL and log-variance divergence-based training in Appendix A.6.10. Please let us know if you have further questions.
>
> Thank you also for your suggestion to add the log-variance loss to Algorithm 1. However, note that this algorithm only describes the sampling and not the training phase and is therefore independent of the loss. We have previously presented the training algorithms in our appendix and referenced them from the main part. However, we agree with your suggestion and moved the algorithm to the main part; see Algorithm 2 in the revised version (highlighted in green). Moreover, note that the training scheme with replay buffers can be found in Algorithm 4 in the appendix. Please let us know if anything is unclear or if you have further suggestions to improve the writing.
>
>
> ## Off-policy training
>
> We agree that the term *off-policy training* originates from the reinforcement learning community. However, several recent works, e.g., [2, 3, 4], have adopted this term in the same context to denote training the sampler with modified trajectory distributions (such as by using replay buffers). As such, we consider *off-policy training* to be established terminology. Nevertheless, we agree that the term should have been better clarified due to its relative novelty in this context, and thank the reviewer for bringing this to our attention. We have added Remark A.3 in our revised paper regarding this and referenced it in Section 2.3 (highlighted in green).
>
>
> ## Table 1 and Differences from alternative methods
>
> Thank you for your suggestions on the writing of this part, which has helped us improve our paper. We, for instance, added lines 125-126 and referenced Appendix A.1 in our Table 1. Due to the large amount of related literature present on sampling problems, SMC methods, and probabilistic modeling, we chose to defer discussion on specific methods and their differences to SCLD to the appendix, mainly due to space limitations in the main part of the paper. We hope that our "related works" section in Appendix A.1 addresses the author's concerns and we also significantly extended it in our new revision. As for PDDS, it is also due to our lack of space that the discussion was deferred to the appendix. However, we now added an extensive comparison between SCLD and PDDS that demonstrates the superiority of SCLD on our set of benchmark tasks (see Appendix A.6.7, highlighted in blue).
>
> We hope that the added content proves satisfactory, and would be happy to work further with you to improve our writing quality.
>
> (response continued in next post)

---

> > ### Author Response · Authors · 2024-11-20
> > **Response to reviewer TwR8 (cont)**
> >
> > ## Finite-time convergence of SCLD
> >
> > Thank you for asking about the finite-time convergence of SCLD, which is indeed a crucial part of our method. In principle, the idea of diffusion-based sampling is to learn the SDE used for sampling in such a way that we reach sampling from the target distribution after finite time -- in contrast to MCMC-based algorithms, where convergence typically appears only asymptotically, so after infinite time. As shown in [5], the CMCD algorithm, on which our method builds, reaches the target distribution exactly if $u=u^*$ in the SDE (6), where $u^*$ is the optimal control that minimizes the loss (11) (based either on the KL or the log-variance divergence). In fact, if $u=u^*$, then all the SMC weights are equal and resampling is in principle not necessary anymore. So in some sense combining diffusion-based sampling with resampling can be motivated by 1) making training more efficient and 2) making sampling more robust if $u \neq u^*$ and these benefits are reflected in our empirical experiments. Please let us know in case you have further questions.
> >
> > ## Benefits of continuous time framework
> >
> > From a theoretical perspective, a continuous time framework allows us to carry forward the results from the CMCD paper such as the existence and uniqueness proof (proposition 3.2 in [5]) which require a continuous-time setting.
> >
> > From a practical perspective, the continuous-time framework gives us more flexibility, by allowing us to decouple design choices regarding what time discretization or integrator to use from the SMC design choices. For instance, we can in principle choose arbitrary points in time at which we apply SMC ingredients (resampling and MCMC refinements). It is important to note that CMCD does not contain these ingredients at all and it is our achievement to add them and show how to still obtain a principled training objective. Our experiments in Section 3 clearly show that adding SMC to CMCD typically improves results significantly and leads to faster and more robust training.
> >
> > ---
> > **References**
> > [1] Nikolas Nüsken and Lorenz Richter. Solving high-dimensional Hamilton–Jacobi–Bellman PDEs using neural networks: perspectives from the theory of controlled diffusions and measures on path space. Partial differential equations and applications, 2(4):48, 2021.
> >
> > [2] Dinghuai Zhang, Ricky Tian Qi Chen, Cheng-Hao Liu, Aaron Courville, and Yoshua Bengio. Diffusion generative flow samplers: Improving learning signals through partial trajectory optimization. arXiv preprint arXiv:2310.02679, 2023a.
> >
> > [3] Sendera, Marcin, et al. "Improved off-policy training of diffusion samplers." The Thirty-Eighth Annual Conference on Neural Information Processing Systems. ACM, 2024.
> >
> > [4] Lorenz Richter and Julius Berner. Improved sampling via learned diffusions. In The Twelfth International Conference on Learning Representations, 2024.
> >
> > [5] Francisco Vargas, Shreyas Padhy, Denis Blessing, and Nikolas Nusken. Transport meets variational inference: Controlled monte carlo diffusions. In The Twelfth International Conference on Learning Representations, 2024.

---

> > > ### Comment · Reviewer_TwR8 · 2024-11-22
> > > **Official Comment by Reviewer TwR8**
> > >
> > > I appreciate the authors' detailed response. The motivation and presentation of the work is now clear to me. I have accordingly adjusted my scores.

---

> > > > ### Author Response · Authors · 2024-11-23
> > > >
> > > > We are thrilled to hear that the motivation and presentation are now clear and appreciate that you updated your score accordingly. We were wondering if there are any remaining questions or concerns we could answer that influence your opinion about our work?

---

### Official Review · Reviewer_Kjto · 2024-10-31

**Soundness:** 3
**Presentation:** 3
**Contribution:** 2
**Rating:** 6
**Confidence:** 3

**Summary:**

This paper considered the task of sampling from unnormalized densities and proposed a framework that combines the sequential Monte Carlo (SMC) method with diffusion-based samplers. The main idea behind such a framework is to view both SMC and diffusion-based samplers under the continuum time limit and consider measures on path spaces. This yields the Sequential Controlled Langevin Diffusion (SCLD) method, which has achieved competitive performance on multiple real-world and synthetic examples.

**Strengths:**

This paper is presented clearly and in detail, making it easy for readers to follow. The proposed framework also allows the authors to design suitable loss functions compatible with off-policy training, which has greatly reduced the use of the training budget in practice. Extensive numerical experiments are provided to validate the effectiveness of the SCLD method.

**Weaknesses:**

1. As the SCLD algorithm relies on the diffusion bridge formulation, the reviewer thinks that it might be necessary to include a short literature review on related work [1,2,3] combining diffusion bridges/stochastic optimal control with generative models for the sake of completeness. However, this is missing in the current version of the manuscript.

2. Though the framework based on path measures proposed in this paper leads to effective sampling methods, the idea of combining SMC with diffusion-based samplers isn't new. In fact, this has been explored in earlier work [4] for the posterior sampling problem, which the authors should have cited and discussed. Specific concerns about the comparison between the SCLD method and [4] will be further elaborated in the "Questions" section below.

**Questions:**

The reviewer's main concern is that a thorough comparison between the SCLD method and the method proposed in [4], whose training stage is based on score matching, needs to be included. Could the authors provide some intuition on how the training losses based on diffusion bridges compare to the score-matching loss when combined with SMC? Empirically, it would be necessary to add the algorithm in [4] as a baseline and compare it with SCLD.

References:

[1] Shi, Y., De Bortoli, V., Campbell, A. and Doucet, A., 2024. Diffusion SchrÃ¶dinger bridge matching. Advances in Neural Information Processing Systems, 36.

[2] De Bortoli, V., Thornton, J., Heng, J. and Doucet, A., 2021. Diffusion schrÃ¶dinger bridge with applications to score-based generative modeling. Advances in Neural Information Processing Systems, 34, pp.17695-17709.

[3] Domingo-Enrich, C., Han, J., Amos, B., Bruna, J. and Chen, R.T., 2023. Stochastic optimal control matching. arXiv preprint arXiv:2312.02027.

[4] Wu, L., Trippe, B., Naesseth, C., Blei, D. and Cunningham, J.P., 2024. Practical and asymptotically exact conditional sampling in diffusion models. Advances in Neural Information Processing Systems, 36.

---

> ### Author Response · Authors · 2024-11-20
> **Response to reviewer Kjto**
>
> Dear Reviewer Kjto,
>
> Thank you for your helpful review and for appreciating our clear and detailed writing as well as the effectiveness of our proposed method. We comment on your questions and concerns in the following:
>
>
> ## Extended literature review
>
> Thank you for the suggestion. Since our combined related work sections have already been quite long and generative modeling focuses on a different task, we did not mention these approaches in our submission. However, we agree that we should discuss the most related approaches and have added a detailed discussion in Appendix A.1 (highlighted in brown), also covering (generalized) Schrödinger bridges. Based on your suggestion, we also added another section on posterior sampling with diffusion-based models and connections to stochastic optimal control.
>
> ## Posterior sampling with diffusion models (and SMC)
>
> Since posterior sampling is a special case of the sampling problem, it can be directly tackled with SCLD without reliance on any samples. However, for complex and very high-dimensional prior distributions, e.g., image/audio/video distributions, it becomes crucial to leverage data.
> In particular, most popular diffusion-based sampling approaches leverage a pre-trained diffusion prior, i.e., a diffusion model trained on samples from the prior distribution. During inference, i.e., the generative process, additional guidance terms are then approximated on the fly to obtain approximate samples from the posterior. This is also the setting of [4], which additionally leverages SMC to obtain asymptotic (in the number of particles) unbiased samples. We present details in Appendix A.1 (highlighted in green).
>
> However, in our setting, we do not have a pretrained diffusion prior (and for many of our tasks not even a prior distribution in the Bayesian sense). Without a pretrained diffusion prior, the method of [4] cannot be applied or (when using an uninformed prior) is effectively just a version of SMC, against which we already compare in our experiments. Note that we also added a comparison in Appendix A.6.8 (highlighted in orange) showing that SCLD can also outperform more advanced SMC methods. The most related method leveraging diffusion-based samplers and SMC is PDDS; see Appendix A.1 for a description and comparison. However, SCLD offers several advantages, leading to better performance (see Appendix A.6.7, highlighted in blue).
>
> Nevertheless, we believe that (non-trivial) adaptations of SCLD are possible to apply its main idea to posterior sampling problems and reward fine-tuning of diffusion models. We mention an outlook at the end of Appendix A.1 and thank the reviewer for bringing up this exciting avenue for future research.

---

> ### Author Response · Authors · 2024-11-30
> **Followup to response**
>
> Dear Reviewer Kjto,
>
> We hope this message finds you well. As the review period is coming to an end, we wanted to know if you have further questions or concerns which we can address?

---

### Official Review · Reviewer_6QXs · 2024-11-04

**Soundness:** 3
**Presentation:** 3
**Contribution:** 3
**Rating:** 8
**Confidence:** 3

**Summary:**

Sequential Monte Carlo and diffusion-based samplers both intend to simulate samples from an un-normalized target distribution via a continuously-evolving sequence of distributions, yet their drawbacks are markedly different: SMC requires no learning but lacks flexibility, which might lead to slow convergence; diffusion samplers are potentially more adaptive but it requires training.

This paper proposes sequential controlled Langevin diffusion (SCLD), a combination of the above two methods that uses diffusion-based samplers to adaptive transition between different stages of SMC. As a result from Girsanov's theorem, we can use the path measure likelihood ratio to construct a sequence of importance sampling weights, which can be used as part of resampling for the SMC procedure. The paper illustrates the results with theoretical and empirical findings that showcase SCLD as a strong alternative to SMC-based samplers.

**Strengths:**

The paper is well-written and I find the results to be of significant interest to the Bayesian statistics community -- SMC has become an important statistical tools for simulating samples from intractable densities, and I think merging SMC with diffusion gives useful insights in building more flexible SMC samplers.

While some of the paper's contributions seem incremental, the resulting algorithm seems more useful in practice when compared to its predecessor work CMCD, mainly thanks to its use of SMC resampling techniques for more accurate sampling.

**Weaknesses:**

I have a positive assessment of this paper, and I enjoyed reading it overall.

The overall contribution of this paper seems somewhat limited. It is somewhat difficult to pinpoint exactly what is new from this paper, although I think the overall readability and SCLD's potential usefulness make up for this weakness.

I think the main weakness of this paper is that the combination of continuous and discrete time seems somewhat arbitrary. It seems to me that given enough training iterations, SCLD or CMCD will be able to learn the optimal control function eventually. In that case there would be no need for resampling and MCMC steps as a well-calibrated SDE solver would be able to simulate a sample from 0 to T directly. Therefore, the main appeal of SCLD is that it is able to correct small imperfections of the control function via resampling (as evidenced empirically by SCLD requiring fewer iterations to achieve good performance). However, when to resample depends crucially on where the discrete $t_n$s are placed. This framework has some flexibility in this regard in that it learns a smooth transition function, but the question of how many discrete steps to take remains.

**Questions:**

- Equation 12: the right hand side looks like the log-density ratio instead of the density ratio itself -- is this correct?
- Figure 5 seems to suggest that the more SMC steps, the better the performance, even though the paper noted potential downsides of excessive resampling (line 521). Have you tried applying an unusually large number of resampling steps, and seen if the results would degrade overall?

---

> ### Author Response · Authors · 2024-11-20
> **Response to reviewer 6QXs**
>
> Dear Reviewer 6QXs,
>
> Thank you very much for your careful review. We are very pleased that you enjoyed reading our paper and found it a useful contribution. We would like to address your concerns below.
>
> ## Overall contributions
> We provide several novel ideas in our paper. We introduce a new SMC framework for combining SMC with diffusion samplers by adopting a continuous path space perspective to SMC and formulating a novel family of training objectives, culminating in the highly effective Sequential Controlled Langevin Diffusion (SCLD) method. Please refer to our General Response for a more detailed discussion of our contributions.
>
> ## The combination of continuous and discrete time seems somewhat arbitrary
>
> Overall, we believe that our framework is a principled way to combine continuous-time diffusion samplers and traditionally discrete-time SMC approaches to take advantage of the strengths of both methods. We agree with the reviewer that if a perfect control is learned then resampling wouldn't be needed, and that when the learned control isn't perfect, SCLD can correct for "imperfections of the control function via resampling". This is a key benefit of our method, leading to significantly improved convergence and performance.
>
> The number of $t_n$'s, as the reviewer points out, is an important design choice in SCLD, and one which we study in Section 3.4 (the part on the choice of number of SMC steps). This must be determined by the available computational budget (a larger number of discretization steps take longer to simulate) as well as the specific target at hand. We found that on most tasks using large numbers of resampling steps (for a fixed total discretization step count) worked well and we did not optimize the position of the $t_n$ to have a fair comparison to our baselines. However, it is possible that performance could be further improved by using more elaborate strategies: as mentioned in the discussion with Reviewer ZuHY, the number of resampling steps and their locations can also be chosen systematically and adaptively as in [3], and this constitutes an area of future work.
>
> We will further discuss the choice of the number of $t_n$'s in your subsequent question regarding the downsides of excessive resampling.
>
> ## Equation 12
> Thank you for identifying our typo, we have amended Equation 12 (in red).
>
> ## Figure 5 seems to suggest that the more SMC steps, the better the performance, even though the paper noted potential downsides of excessive resampling (line 521).
> Yes, we have found that excessive resampling can impact performance on highly multimodal tasks. In Figure 4 (previously Figure 5), you can see that on the Robot4 task, the method performs worse when 128 resampling steps are performed at training and evaluation, compared to using 4 resampling steps at training and none at evaluation. This is the result of mode collapse. We also found that in the GMM40 (50d) task, using 128 resampling steps led SCLD to collapse onto one mode. As such, we used fewer (4) resampling steps on multimodal tasks as opposed to 128. However, our results were all obtained using *multinomial resampling*, which is well known to be suboptimal [1], in the interests of giving a fair comparison to methods like CRAFT. It is likely that mode collapse and loss of sample diversity can be mitigated by using better resampling schemes such as systematic or parallel resampling [2].
>
>
>
> **References**
>
> [1] https://arxiv.org/pdf/2402.06320
>
> [2] https://arxiv.org/abs/1903.12583
>
> [3] https://arxiv.org/abs/1808.07730

---

> > ### Comment · Reviewer_6QXs · 2024-11-26
> >
> > Thank you very much for your response. I think some further improvements have been made to the paper, and I will maintain my assessment as I already provide a positive review for this paper.

---

### Author Response · Authors · 2024-11-20
**General response**

We would like to thank all reviewers for their thoughtful comments and insights, which have significantly strengthened the paper overall. We have now uploaded a revised version of the manuscript, which we hope reflects the advice provided by the reviewers.

We are delighted to hear that the reviewers found our contributions to be a "novel and promising approach" (reviewer TwR8) and "the results to be of significant interest" (reviewer 6QXs).

## Summary of rebuttal

We will now outline our main changes to the paper. For more details, please see the individual responses posted to each reviewer.

1. **Improved presentation:** Thanks to the suggestion of all reviewers, we were able to enact various changes to improve our writing quality. This includes adding Algorithm 2 to the main body of the paper, which describes the basic training idea of SCLD (as suggested by reviewer TwR8), and clarifications to various statements.
2. **Additional experiments and baselines:** In response to general queries, we have added two detailed comparisons with relevant baselines. In section Appendix A6.7, we provide a comparison to Particle Denoising Diffusion Samplers (PDDS) [1], as well as a discussion of learning a prior for SCLD using variational inference in Appendix A6.6. In section Appendix A6.8 we provide a comparison to a more sophisticated SMC scheme in [2] as recommended by reviewer ZuHY. Finally, we mention promising first results on the $\phi^4$-lattice field theory in the response to reviewer ZuHY.
3. **Theoretical analysis of log-variance loss:** We added a theoretical analysis of the favorable scaling behavior of the relative error of the log-variance divergence in Appendix A.2 as suggested by reviewer TwR8.
4. **Extended related works:** We have greatly expanded our related works section in Appendix A.1 to include a literature review on posterior sampling and diffusion-based generative modeling, as well as other methods in response to the suggestions of reviewers Kjto and TwR8
5. **Additional visualizations:** We added a visual comparison of the learned annealing schedule in Appendix A6.5 with a linear annealing schedule (in orange).

To indicate clearly what has changed, we colour-code the modifications according to the reviewer who suggested it as follows:
- red: reviewer 6QXs
- brown: reviewer Kjto
- olive (yellow-green): reviewer TwR8
- orange: reviewer ZuHY
- other changes: blue


## Summary of our contributions

We would also like to reiterate our core contributions, as raised by multiple authors:
- **General SMC framework in path space:** We provide a novel general framework for combining diffusion-based samplers with SMC by constructing a continuous time version of SMC that performs importance sampling in path space. This allows great flexibility in independently selecting the times for resampling and MCMC during training and inference (see Fig. 4).

- **Principled objectives:** We introduce a novel family of training losses for this general framework (equation 14) that enables the sampler to be trained end-to-end (i.e., "directly" in the same fashion as it is used during inference). We explain, with theoretical justification, why a member of this family, based on the log-variance divergence (equation 18), is particularly desirable. To the best of our knowledge, this is the first objective for SMC-based samplers that (1) can be trained end-to-end including hyperparameters, (2) does not require importance sampling (exhibiting potentially high variance, see Prop. 2.2), and (3) allows for a principled use of replay buffers.

 - **SOTA performance:** Building on those connections and adapting ideas from CMCD, we propose our new sampling method Sequential Controlled Langevin Diffusion (SCLD) as a special case of our framework, and perform extensive experimental validation on it. SCLD achieves state-of-the-art performance across a wide range of benchmarks and metrics with significantly reduced training costs.

We hope that we have addressed most of the concerns raised by the reviewers. We would be very grateful for additional feedback.

---

### Meta-Review · Area_Chair_Ei7W · 2024-12-20

**Metareview:**

The paper introduces Sequential Controlled Langevin Diffusions (SCLD), a novel algorithm that combines Sequential Monte Carlo (SMC) and diffusion-based sampling to efficiently sample from unnormalized target distributions, achieving state-of-the-art performance on various benchmarks. The paper presents a novel and theoretically grounded framework for combining SMC and diffusion models, leading to a practical algorithm with improved efficiency and performance compared to existing methods. While the core idea is strong, some reviewers found the initial presentation lacked clarity and questioned the necessity of combining continuous and discrete time elements, prompting the authors to provide further clarification and experimental justification. The reviewers are overall in favor of accepting this paper.

**Additional Comments On Reviewer Discussion:**

Despite some initial concerns about clarity and justification, the authors adequately addressed these issues through revisions and additional experiments, demonstrating the value and novelty of their proposed SCLD framework.

---

### Decision · Program_Chairs · 2025-01-22

Accept (Poster)